# QSCA: Quantization with Self-Compensating Auxiliary for Monocular Depth Estimation

**Jincheol Yang**[*]
Department of Electronic Engineering
Sogang University
yjc3232@sogang.ac.kr

**Jaemin Choi**[*]
Department of Electronic Engineering
Sogang University
jam0225@sogang.ac.kr

**Matti Zinke**[*]
Department of Computer Science
Sogang University
mattizinke@sogang.ac.kr

**Suk-Ju Kang**
Department of Electronic Engineering
Sogang University
sjkang@sogang.ac.kr

## Abstract

Monocular depth estimation has advanced significantly with foundation models like Depth Anything, leveraging large-scale transformer architectures for the superior generalization. However, the deployment on resource-constrained devices remains challenging due to the high computation and memory requirement. Existing quantization methods, such as post-training quantization (PTQ) and quantization-aware training (QAT), often face trade-offs between efficiency and accuracy, or require extensive labeled data for retraining. To address these limitations, we propose Quantization with Self-Compensating Auxiliary for Monocular Depth Estimation (QSCA), a novel framework for 4-bit post-training quantization of Monocular depth estimation models. Our method integrates a lightweight Self-Compensating Auxiliary (SCA) module into both transformer encoder and decoder blocks, enabling the quantized model to recover from performance degradation without requiring ground truth. This design enables fast adaptation while preserving structural and spatial consistency in predicted depth maps. To our knowledge, this is the first framework to successfully apply 4-bit quantization across all layers of large-scale monocular depth estimation models. Experimental results demonstrate that QSCA significantly improves quantized depth estimation performance. On the NYUv2 dataset, it achieves an 11% improvement in $\delta_1$ accuracy over existing post-training quantization methods.

## 1 Introduction

Monocular depth estimation (MDE) has become a fundamental task of modern computer vision, in robotics [1], autonomous driving [2, 3], and 3D scene understanding [4, 5]. The recent emergence of foundation models such as MiDaS [6], Depth Anything v1 [7], and Depth Anything v2 [8] has dramatically improved the accuracy and generalization of depth prediction by leveraging transformer-based architectures along with large-scale datasets, both labeled and unlabeled. These advances have shifted the paradigm from the hand-crafted feature engineering to the data-driven, generalizable depth estimation. Despite their impressive capabilities, deploying these large-scale models on resource-constrained platforms remains a significant challenge. Real-world scenarios such as edge devices demand models that are highly resource-efficient.

---

[*]Equal contribution

39th Conference on Neural Information Processing Systems (NeurIPS 2025).

To address this issue, a range of advanced model compression strategies have been explored, including pruning [9, 10], knowledge distillation [11, 12], model quantization [13–20], and optimized architecture design [21–24], with the objective of reducing model complexity while maintaining performance. Model quantization, especially post-training quantization (PTQ) [25–27], has emerged as a practical solution to reduce model size and accelerate inference without requiring retraining. PTQ methods seek quantization parameters using a few unlabeled calibration images after full-precision training, offering rapid deployment and minimal data requirements. However, PTQ struggles under aggressive low-bit quantization (e.g., 4-bit), suffering from substantial accuracy degradation due to quantization errors, sensitivity to outliers, and the mismatch between quantized representations and real data distribution. These issues are amplified in complex tasks like depth estimation, where fine-grained spatial details are critical.

Quantization-aware training (QAT) [28–30] addresses these limitations by simulating quantization effects during training, allowing the model to adapt its weights for robustness against quantization error. QAT consistently outperforms PTQ under low-bit quantization by adapting model parameters to quantization noise during training. However, it imposes significant computational and memory overhead, and typically requires access to labeled datasets, a condition that often does not hold for foundation models trained on proprietary or large-scale unlabeled data. QwT [31] achieves a balance of speed, accuracy, and simplicity by introducing lightweight compensation modules that recover information lost by quantization. These modules can be optimized efficiently via a closed-form solution, enabling rapid adaptation without full retraining. While QwT effectively bridges the gap between PTQ and QAT by enabling lightweight supervised adaptation, it still relies on labeled data during the training of its compensation modules. As a result, QwT is not well suited for foundation monocular depth estimation models, which are typically evaluated in zero-shot configurations where precise depth prediction is required but ground-truth labels are unavailable.

This paper proposes Quantization with Self-Compensating Auxiliary for Monocular Depth Estimation (QSCA), a novel framework for efficient 4-bit post-training quantization of foundation MDE models. As illustrated in Figure 1, our framework introduces lightweight Self-Compensating Auxiliary (SCA) modules into transformer and decoder blocks. These modules are inserted into quantized blocks to mitigate degradation by restoring critical features via residual correction. The SCA modules are trained via self-supervised learning, enabling the model to recover quantization induced loss using only unlabeled calibration data, without reliance on ground-truth depth labels. This approach dramatically reduces adaptation time and computational requirements compared to traditional QAT or reconstruction-based PTQ methods, while maintaining high accuracy even with aggressive quantization. Our main contributions are as follows:

- We propose the QSCA framework, the first to effectively apply 4-bit quantization across all layers of large-scale monocular depth estimation foundation models such as Depth Anything.
- We propose SCA, a set of lightweight auxiliary modules strategically integrated into the quantized network to restore lost representational capacity. These modules are trained using self-supervised learning, eliminating the need for any ground-truth depth annotations and enabling fast quantization adaptation.
- We demonstrate that our framework achieves competitive performance on relative depth prediction across multiple benchmark datasets under a 4-bit quantization setting, significantly outperforming existing PTQ baselines.

## 2 Related Work

### 2.1 Monocular Depth Estimation Models

MDE has made significant progress with the emergence of large-scale pretrained models, often referred to as foundation models. Notable examples include MiDaS [6], Depth Anything v1 [7], and Depth Anything v2 [8], which leveraged large-scale labeled and unlabeled datasets alongside transformer-based architectures to achieve high generalization performance across a wide range of scenes. MiDaS introduced cross-dataset generalization through a mixture of 12 diverse datasets and a transformer encoder-decoder design based on DPT [32]. Depth Anything extended this framework using a DINOv2 [33] ViT encoder and DPT decoder, with v1 leveraging labeled and unlabeled data, and v2 focusing on synthetic and pseudo-labeled data. Both were evaluated in zero-shot settings and became strong baselines for general-purpose depth estimation. MiDaS and both versions of

Depth Anything were built upon DPT [32], which adopted a ViT-based encoder-decoder architecture specifically designed for dense prediction tasks. DPT captures long-range dependencies through a transformer encoder and recovers spatial details via a multi-scale decoder, making it well-suited for monocular depth estimation.

## 2.2 Model Quantization

Model quantization [13–15] aims to improve computational efficiency by reducing the precision of floating point weights and activations. AdaRound [20] proposed a layer-wise reconstruction approach that refines the rounding direction of weights in each quantized layer to minimize task loss. BRECQ [17] adopted block-wise reconstruction to account for cross-layer dependencies. QDrop [18] incorporated randomly dropping the quantization of activations during block-wise reconstruction, effectively applying partial activation quantization in a stochastic manner. Although these methods demonstrated strong performance in classification tasks, their design was specifically optimized for cross-entropy loss [17, 34, 35]. As a result, their effectiveness was significantly diminished when applied to tasks that required alternative loss functions, and the reconstruction process further incurred substantial resource costs. RepQ-ViT [36] introduced scale reparameterization, which decouples the quantization and inference processes for post-LayerNorm and post-Softmax in ViTs [37–39]. AdaLog [19] designed a hardware-friendly adaptive logarithmic quantizer to address the power-law distributions in ViTs. These methods offered the advantage of faster results by eliminating the need for reconstruction. However, since these approaches was primarily designed for image classification [37–39], they suffered from substantial performance degradation when applied to monocular depth estimation task. Inspired by QwT [31], which introduced a compensation module to recover performance degradation, we propose a more robust quantization methodology tailored for monocular depth estimation.

## 3 Method

We propose the QSCA framework, which introduces SCA modules in the network to recover critical depth representations degraded by quantization. As illustrated in Figure 1, we incorporate these modules into both the Q-Transformer and Q-DPT blocks to effectively mitigate quantization loss. To optimize the SCA modules, we employ a self-supervised learning process that distills knowledge from the full-precision (FP) model by aligning intermediate features and predictions between the quantized and FP models. Section 3.3 provides a detailed description of the SCA module design, and Section 3.4 focuses on the self-supervised learning strategy used to optimize the SCA without requiring ground-truth annotations.

### 3.1 Preliminaries

**Uniform quantization.** We follow the uniform quantization used in the ViT quantization method [27, 40, 41]. In the uniform quantization process, given the full-precision value $x$ and the bit-width $b$, the quantized value $\hat{x}$ is computed as follows:

$$\bar{x} = \text{clip}\left(\left\lfloor \frac{x}{s} \right\rceil + z, 0, 2^b - 1\right), \quad \hat{x} = s \cdot \bar{x}, \tag{1}$$

where $\lfloor \cdot \rceil$ denotes the round function and the clip function constrains the input value within a given upper and lower bound. In this context, the value is constrained between 0 and $2^b - 1$. The scale factor $s$ maps the original data range to the range that can be represented by $2^b$ levels. The zero-point $z$ serves to shift the quantized range to adequately represent both positive and negative values. The $s$ and $z$ are defined as:

$$s = \frac{\max(x) - \min(x)}{2^b - 1}, \quad z = \left\lfloor \frac{-\min(x)}{s} \right\rceil. \tag{2}$$

**Log$\sqrt{2}$ Quantization.** To better handle long-tailed distributions such as post-Softmax activations, RepQ-ViT [36] introduces Log$\sqrt{2}$ quantization, which provides finer granularity for larger values compared to Log2 quantization. This formulation offers higher resolution for large-magnitude values, which helps preserve rank order in attention scores. The quantized value $\hat{x}$ is computed as follows:

$$\bar{x} = \text{clip}\left(\left\lfloor -\log_{\sqrt{2}} \frac{x}{s} \right\rceil, 0, 2^b - 1\right), \quad \hat{x} = s \cdot 2^{\left\lfloor -\frac{\bar{x}}{2} \right\rfloor} \cdot [\mathbf{P}(\bar{x}) \cdot (\sqrt{2} - 1) + 1], \tag{3}$$

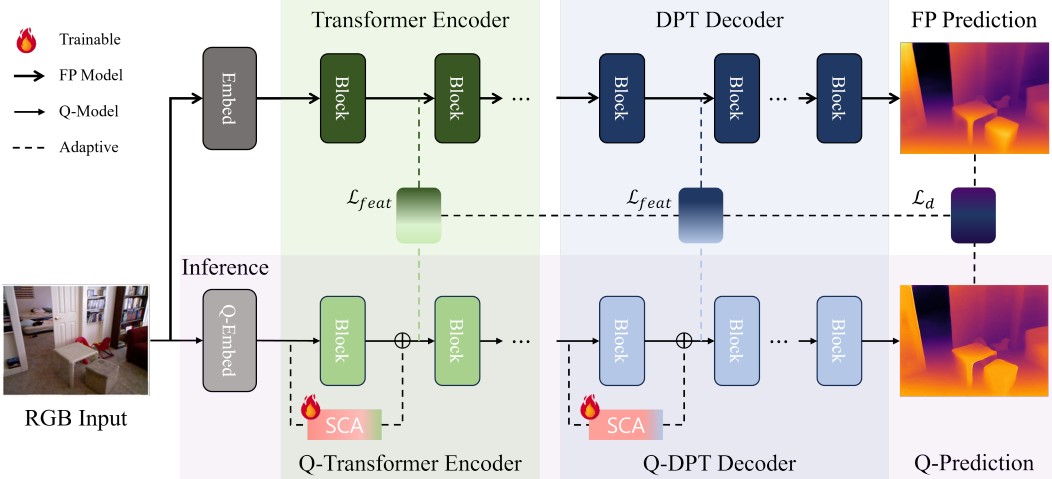

Figure 1: Overview of the QSCA framework. In the quantized Q-model, SCA modules are strategically inserted into each block of both the Transformer encoder (Q-Transformer) and the DPT decoder (Q-DPT) to compensate for performance degradation caused by quantization. These SCA modules are trained in a self-supervised manner by distilling intermediate features and predictions from the full-precision (FP) model at each corresponding block.

where $\lfloor \cdot \rfloor$ denotes the floor function and $\mathbf{P}(\cdot)$ is a function that returns 0 if input is even and 1 if it is odd.

## 3.2 Bit-level and Block-wise Quantization Sensitivity

To better understand the quantization behavior of large-scale MDE models, we analyze the performance degradation of the Depth Anything [7] applying percentile-based PTQ approaches. The calibration is performed using 16 randomly selected samples from the train dataset. The results presented in Figure 2 offer critical insight into the quantization robustness of the Depth Anything model under 4-bit post-training quantization. In Figure 2 (a), we observe a steep degradation in performance metrics when both weights and activations are quantized to 4-bit precision. Specifically, the $\delta_1$ metric drops consistently while AbsRel increases, demonstrating that aggressive bit-level compression introduces substantial information loss. This degradation pattern is consistent across both NYUv2 [42] and KITTI [43] benchmarks, indicating that existing PTQ methods fail to maintain predictive reliability under aggressive compression in high-resolution depth estimation tasks. To further investigate the vulnerability of individual components in the model, we perform a block-wise quantization sensitivity analysis across the blocks of the model, as shown in Figure 2 (b). Each block in the architecture is independently quantized to 4-bit precision while keeping the rest in full precision, and the drop in $\delta_1$ is measured. While most blocks exhibit moderate sensitivity to quantization, we observe that the decoder, particularly the final CNN-based block, experiences the most significant performance degradation. This indicates that the decoder plays a crucial role in maintaining spatial and structural fidelity and is especially vulnerable under low-bit quantization.

## 3.3 Self-Compensating Auxiliary Module

We implement SCA with linear layer in transformer blocks and convolutional layers in decoder blocks. Each SCA module operates in a residual manner by taking the input to the quantized block and adding the auxiliary output to the quantized output of the block. This avoids reconstructing or modifying quantized weights.

Formally, let $\hat{x} \in \mathbb{R}^{d_{in}}$ be the input activation to a quantized block, and let $\hat{o} \in \mathbb{R}^{d_{\text{out}}}$ denote the corresponding quantized output. We define the compensated output as:

$$o^{\text{SCA}} = \hat{o} + \phi(\hat{x}; \theta), \qquad (4)$$

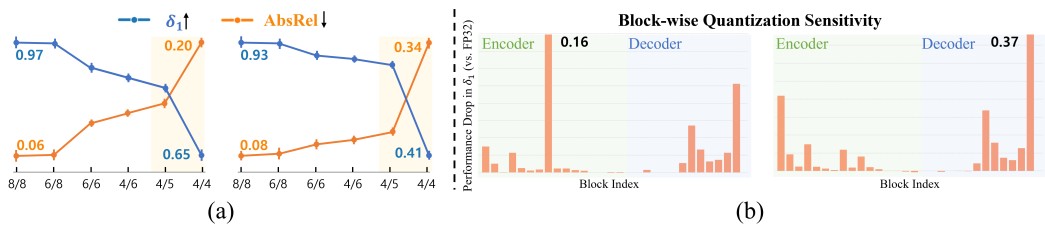

Figure 2: Quantization sensitivity analysis of the Depth Anything small model on NYUv2 and KITTI datasets. (a) Performance degradation when both weights and activations (W/A) are quantized to lower bit-widths. (b) Block-wise sensitivity to 4-bit quantization, where a higher bar indicates that the corresponding block is more sensitive to quantization.

where $\phi(\hat{x}; \theta)$ is the output of the SCA module, implemented as a linear layer in the Q-Transformer encoder or a convolutional layer in the Q-DPT decoder. To initialize the parameters $\theta$ of the auxiliary module, we treat $\phi$ as a linear projection from the quantized input representation to the residual correction target. Let $\hat{\mathbf{X}} \in \mathbb{R}^{(B \times L) \times d_{in}}$ denote the reshaped input tokens, and let the residual matrix be defined as $\mathbf{R} = \mathbf{O} - \hat{\mathbf{O}}$, where $\mathbf{O}$ and $\hat{\mathbf{O}}$ denote the full-precision and corresponding quantized outputs, respectively.

To ensure stable and robust initialization, we incorporate an L2 regularization term into the objective, which penalizes the Frobenius norm of the projection matrix. The optimal projection matrix $\mathbf{W}\phi^{\star} \in \mathbb{R}^{d_{out} \times d_{in}}$ is thus computed as:

$$\mathbf{W}_{\phi}^{\star} = \arg \min_{\mathbf{W}} \|\mathbf{R} - \hat{\mathbf{X}}\mathbf{W}^{\top}\|_F^2 + \lambda_1 \|\mathbf{W}\|_F^2, \tag{5}$$

where $\lambda_1$ controls the strength of the regularization. $\lambda_1$ ensures that $(\hat{\mathbf{X}}^{\top}\hat{\mathbf{X}} + \lambda_1\mathbf{I})$ is always invertible, thereby enhancing the stability of weight initialization by guaranteeing the existence of a unique and numerically stable solution for the projection matrix. By differentiating the objective in Eq. (5) with respect to $\mathbf{W}$ and setting the derivative to zero, we obtain the following closed-form solution for the optimal projection matrix:

$$\mathbf{W}_{\phi}^{\star} = (\hat{\mathbf{X}}^{\top}\hat{\mathbf{X}} + \lambda_1\mathbf{I})^{-1}\hat{\mathbf{X}}^{\top}\mathbf{R}. \tag{6}$$

Consequently, the output of the quantized block with the SCA module is thus computed as:

$$\mathbf{O}^{\mathbf{SCA}} = \hat{\mathbf{O}} + \hat{\mathbf{X}}\mathbf{W}_{\phi}^{\star\top}. \tag{7}$$

This equation implies that the SCA module explicitly compensates for the residual between the original output and the quantized output of the block, effectively restoring performance degraded by quantization.

### 3.4 Self-Supervised Learning Strategy

We adopt a self-supervised learning strategy in which the full-precision model $\mathcal{F}$ acts as the teacher, and the quantized model $\mathcal{F}_q$ augmented with SCA modules serves as the student. The output of $\mathcal{F}$ is treated as pseudo ground-truth, and the SCA parameters in the student are trained to imitate it through a distillation loss. Rather than applying the loss only at the final prediction stage, we introduce auxiliary supervision directly at intermediate blocks where the SCA modules are integrated. Let $\hat{f}_i$ and $f_i$ denote the feature output from the student with SCA and the corresponding output from the teacher at the $i$-th block, respectively. We define the feature-level distillation loss as:

$$\mathcal{L}_{\text{feat}} = \sum_{i \in \mathcal{S}} \|f_i - \hat{f}_i\|_1, \tag{8}$$

where $\mathcal{S}$ is the set of blocks equipped with the SCA modules. In addition, to enforce scale-invariant consistency in relative depth estimation, we adopt a SILog loss between the final depth maps $\hat{D}_q$ from the student and $D$ from the teacher:

$$\mathcal{L}_{\text{d}} = \frac{1}{n}\sum_{j=1}^{n}\left(\log \hat{D}_{q,j} - \log D_j\right)^2 - \frac{1}{n^2}\left(\sum_{j=1}^{n}\log \hat{D}_{q,j} - \log D_j\right)^2, \tag{9}$$

Table 1: Quantization results on the NYUv2 [42] for zero-shot relative depth estimation. W/A indicates the bit-width of weights and activations after quantization. *E.* denotes the encoder backbone used in the MDE architectures.

| Method | W/A | Depth Anything v1 [7] | | | | Depth Anything v2 [8] | | | |
|---|---|---|---|---|---|---|---|---|---|
| | | *E.* ViT-S | | *E.* ViT-B | | *E.* ViT-S | | *E.* ViT-B | |
| | | $\delta_1 \uparrow$ | AbsRel $\downarrow$ | $\delta_1 \uparrow$ | AbsRel $\downarrow$ | $\delta_1 \uparrow$ | AbsRel $\downarrow$ | $\delta_1 \uparrow$ | AbsRel $\downarrow$ |
| FP | 32/32 | 0.9720 | 0.0525 | 0.9791 | 0.0459 | 0.9736 | 0.0513 | 0.9770 | 0.0460 |
| MinMax [14] | 4/4 | 0.5024 | 0.2728 | 0.1972 | 1.5735 | 0.4873 | 0.2815 | 0.1102 | 2.3712 |
| Percentile [44] | 4/4 | 0.6542 | 0.2006 | 0.5430 | 0.2522 | 0.6675 | 0.1935 | 0.5379 | 0.2543 |
| BRECQ [17] | 4/4 | 0.5395 | 0.2535 | 0.4692 | 0.2886 | 0.5042 | 0.2714 | 0.4646 | 0.2910 |
| QDrop [18] | 4/4 | 0.7166 | 0.1742 | 0.5785 | 0.2334 | 0.7115 | 0.1773 | 0.5794 | 0.2332 |
| Ours | 4/4 | **0.8097** | **0.1377** | **0.6845** | **0.1875** | **0.8151** | **0.1361** | **0.6845** | **0.1875** |
| MinMax [14] | 4/6 | 0.5632 | 0.2417 | 0.4973 | 0.2748 | 0.5324 | 0.2599 | 0.4738 | 0.2866 |
| Percentile [44] | 4/6 | 0.8837 | 0.1050 | 0.9071 | 0.0958 | 0.9196 | 0.0891 | 0.9355 | 0.0831 |
| BRECQ [17] | 4/6 | 0.5786 | 0.2337 | 0.5542 | 0.2462 | 0.5269 | 0.2611 | 0.5084 | 0.2697 |
| QDrop [18] | 4/6 | 0.6369 | 0.2071 | 0.6987 | 0.1823 | 0.7285 | 0.1700 | 0.7619 | 0.1560 |
| Ours | 4/6 | **0.9333** | **0.0810** | **0.9441** | **0.0739** | **0.9450** | **0.0757** | **0.9468** | **0.0726** |

Table 2: Quantization results on the KITTI [43] for zero-shot relative depth estimation. W/A indicates the bit-width of weights and activations after quantization. *E.* denotes the encoder backbone used in the MDE architectures.

| Method | W/A | Depth Anything v1 [7] | | | | Depth Anything v2 [8] | | | |
|---|---|---|---|---|---|---|---|---|---|
| | | *E.* ViT-S | | *E.* ViT-B | | *E.* ViT-S | | *E.* ViT-B | |
| | | $\delta_1 \uparrow$ | AbsRel $\downarrow$ | $\delta_1 \uparrow$ | AbsRel $\downarrow$ | $\delta_1 \uparrow$ | AbsRel $\downarrow$ | $\delta_1 \uparrow$ | AbsRel $\downarrow$ |
| FP | 32/32 | 0.9369 | 0.0818 | 0.9396 | 0.0804 | 0.9340 | 0.0832 | 0.9389 | 0.0814 |
| MinMax [14] | 4/4 | 0.3441 | 0.3770 | 0.2058 | 1.9612 | 0.3423 | 0.3938 | 0.0832 | 4.4358 |
| Percentile [44] | 4/4 | 0.4099 | 0.3418 | 0.3327 | 0.3876 | 0.3780 | 0.3668 | 0.3275 | 0.3932 |
| BRECQ [17] | 4/4 | 0.3522 | 0.3719 | 0.3160 | 0.3989 | 0.3344 | 0.3906 | 0.3175 | 0.3990 |
| QDrop [18] | 4/4 | 0.3234 | 0.3934 | 0.3338 | 0.3855 | 0.3748 | 0.3620 | 0.4082 | 0.3412 |
| Ours | 4/4 | **0.7273** | **0.1874** | **0.6203** | **0.2365** | **0.6794** | **0.2067** | **0.6174** | **0.2296** |
| MinMax [14] | 4/6 | 0.4467 | 0.3290 | 0.3586 | 0.3769 | 0.3609 | 0.3744 | 0.3161 | 0.3982 |
| Percentile [44] | 4/6 | 0.8558 | 0.1269 | 0.7580 | 0.1687 | 0.8613 | 0.1283 | 0.7219 | 0.1866 |
| BRECQ [17] | 4/6 | 0.5223 | 0.2851 | 0.4035 | 0.3529 | 0.4320 | 0.3332 | 0.4093 | 0.3497 |
| QDrop [18] | 4/6 | 0.6052 | 0.2408 | 0.5897 | 0.2611 | 0.5351 | 0.2790 | 0.5268 | 0.3037 |
| Ours | 4/6 | **0.8857** | **0.1161** | **0.8722** | **0.1174** | **0.8893** | **0.1124** | **0.8873** | **0.1067** |

where $n$ is the number of valid pixels, and $\hat{D}_q$ is the predicted depth map from the quantized model with SCA. This formulation emphasizes relative depth consistency without relying on absolute scale. Then, the final training objective is given as the sum of the feature distillation loss and the depth-related loss:

$$\mathcal{L}_{\text{total}} = \lambda_{\text{feat}}\mathcal{L}_{\text{feat}} + \mathcal{L}_d. \tag{10}$$

# 4 Experiments

## 4.1 Experimental Setup

**Models, datasets, and metrics.** We adopt Depth Anything v1 [7] and v2 [8], which employ ViT-Small, ViT-Base as backbone encoders [37]. We choose Depth Anything as the baseline model because it represents a strong foundation model for zero-shot monocular depth estimation. Since it is used in zero-shot settings without task-specific fine-tuning, applying quantization enables more efficient deployment in real-world scenarios while preserving generalization capability. For evaluation,

Table 3: Quantization results on the Sintel [45], ETH3D [46], and DIODE [47] for zero-shot relative depth estimation. W/A indicates the bit-width of weights and activations after quantization. *E.* denotes the encoder backbone used in the MDE architectures.

| | Method | W/A | Depth Anything v1 [7] | | | | Depth Anything v2 [8] | | | |
| | | | *E.* ViT-S | | *E.* ViT-B | | *E.* ViT-S | | E. ViT-B | |
| | | | $\delta_1 \uparrow$ | AbsRel $\downarrow$ | $\delta_1 \uparrow$ | AbsRel $\downarrow$ | $\delta_1 \uparrow$ | AbsRel $\downarrow$ | $\delta_1 \uparrow$ | AbsRel $\downarrow$ |
|---|---|---|---|---|---|---|---|---|---|---|
| Sintel | FP | 32/32 | 0.7304 | 0.2296 | 0.7539 | 0.2281 | 0.6974 | 0.2680 | 0.7111 | 0.2576 |
| | MinMax [14] | 4/4 | 0.3217 | 0.4751 | 0.0069 | 54.2156 | 0.3101 | 0.4998 | 0.1727 | 6.1550 |
| | Percentile [44] | 4/4 | 0.3585 | 0.4745 | 0.3654 | **0.4677** | 0.3772 | 0.4693 | 0.3237 | 0.4801 |
| | BRECQ [17] | 4/4 | 0.3123 | 0.4787 | 0.3085 | 0.4820 | 0.3055 | 0.4816 | 0.3008 | 0.4821 |
| | QDrop [18] | 4/4 | 0.3181 | 0.4991 | 0.3106 | 0.4867 | 0.3184 | 0.4948 | 0.3151 | 0.4882 |
| | Ours | 4/4 | **0.3884** | **0.4531** | **0.4059** | 0.4781 | **0.3919** | **0.4474** | **0.4070** | **0.4679** |
| ETH3D | FP | 32/32 | 0.9652 | 0.0584 | 0.9741 | 0.0513 | 0.9701 | 0.0548 | 0.9791 | 0.0467 |
| | MinMax [14] | 4/4 | 0.5264 | 0.2754 | 0.2070 | 3.0313 | 0.5078 | 0.2872 | 0.0833 | 5.4311 |
| | Percentile [44] | 4/4 | 0.6290 | 0.2186 | 0.5935 | 0.2380 | 0.6456 | 0.2148 | 0.5720 | 0.2491 |
| | BRECQ [17] | 4/4 | 0.5082 | 0.2829 | 0.4874 | 0.2958 | 0.4962 | 0.2920 | 0.4870 | 0.2963 |
| | QDrop [18] | 4/4 | 0.6069 | 0.2351 | 0.5373 | 0.2706 | 0.6186 | 0.2288 | 0.5341 | 0.2697 |
| | Ours | 4/4 | **0.7332** | **0.1791** | **0.6309** | **0.2241** | **0.6791** | **0.1983** | **0.6298** | **0.2197** |
| DIODE | FP | 32/32 | 0.9413 | 0.0753 | 0.9474 | 0.0745 | 0.9426 | 0.0721 | 0.9498 | 0.0701 |
| | MinMax [14] | 4/4 | 0.6687 | 0.2110 | 0.1070 | 13.8008 | 0.6596 | 0.2154 | 0.1647 | 11.9492 |
| | Percentile [44] | 4/4 | 0.7347 | 0.1862 | 0.6845 | 0.2038 | 0.7459 | 0.1830 | 0.7027 | 0.2019 |
| | BRECQ [17] | 4/4 | 0.6755 | 0.2079 | 0.6464 | 0.2201 | 0.6538 | 0.2173 | 0.6441 | 0.2211 |
| | QDrop [18] | 4/4 | 0.7275 | 0.1864 | 0.6819 | 0.2058 | 0.7279 | 0.1848 | 0.6777 | 0.2062 |
| | Ours | 4/4 | **0.8033** | **0.1513** | **0.7099** | **0.1997** | **0.8135** | **0.1463** | **0.7094** | **0.1947** |

we employ five widely used monocular depth estimation benchmarks: NYUv2 [42], KITTI [43], Sintel [45], ETH3D [46], and DIODE [47]. These benchmarks cover diverse domains such as indoor environments, outdoor driving scenes, synthetic imagery, and mixed settings. We select them because they correspond to the official zero shot relative depth evaluation datasets used in the Depth Anything models [7, 8]. To quantify model performance, we follow standard evaluation metrics from prior works [6–8, 32], including those used in Depth Anything. We report the accuracy under threshold ($\delta_1 < 1.25^1$) and the absolute relative error (AbsRel). The $\delta_1$ metric indicates the percentage of predicted depth values that lie within a threshold ratio of the ground truth, while AbsRel computes the mean of the absolute relative errors. Higher $\delta_1$ and lower AbsRel values reflect better depth estimation performance.

**Implementation details.** We randomly select 16 samples from each of the NYUv2, and KITTI datasets to calibrate the quantization parameters. For the calibration strategy, we apply the widely used percentile method [44], employing channel-wise quantization for weights and layer-wise quantization for activations. To initialize the parameters of the SCA modules, we randomly sample a single image from the training set. The SCA modules are implemented with FP16 precision to reduce memory usage and model size. When finetuning the SCA modules, we perform 1 epoch of training using only a randomly selected 5% subset of the training set. We adopt this strategy to simulate scenarios with limited data and computational resources. For training, we use the Adam optimizer with a learning rate of $1 \times 10^{-4}$ and without weight decay. The hyperparameters are set as follows: $\lambda_1$ is set to 1e3, and $\lambda_{\text{feat}}$ is set to 0.5. All experiments are conducted with batch size of 1, and a single RTX 4090 GPU.

## 4.2 Experimental Results

**Quantitative results.** As shown in Tables 1 and 2, we evaluate our framework on NYUv2 [42] and KITTI [43] under two quantization settings: 4/4 and 4/6 (weight/activation bit-widths). Our QSCA consistently outperforms prior PTQ methods, including MinMax [14], Percentile [44], BRECQ [17], and QDrop [18]. Specifically, under the more aggressive 4/4 configuration on NYUv2, QSCA applied to Depth Anything v1 achieves a $\delta_1$ of 0.8097 with ViT-S and 0.6845 with ViT-B, outperforming

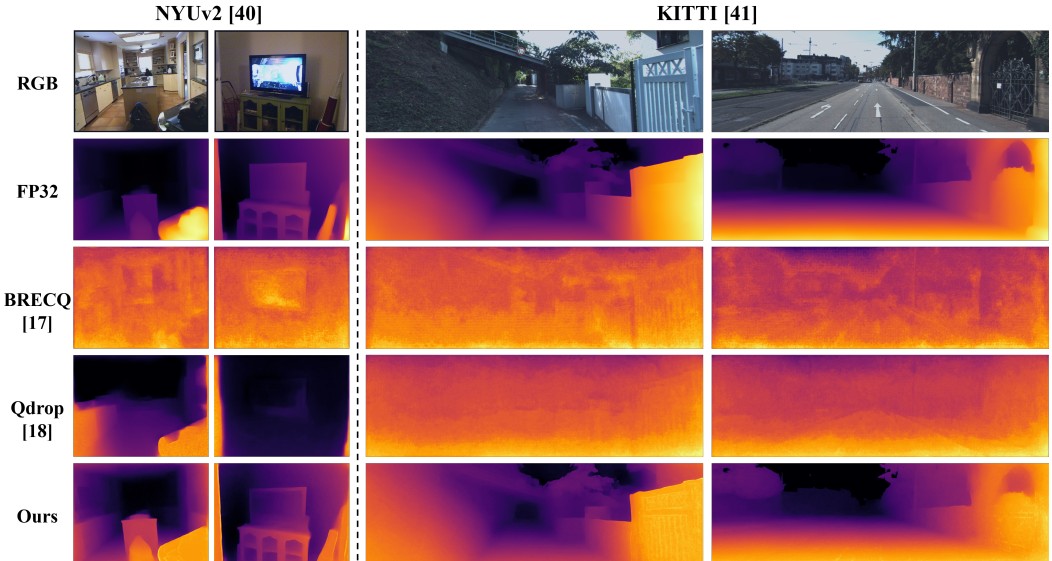

Figure 3: Visualization of the quantized Depth Anything (W4A6) with ViT-small backbone for indoor scenes (NYUv2) and outdoor scenes (KITTI).

Table 4: Efficiency-performance trade-off of QSCA on the NYUv2 using Depth Anything v1 under 4-bit weight and activation quantization.

| Model | Method | Params | Time | $\delta_1 \uparrow$ | AbsRel $\downarrow$ |
|-------|--------|--------|------|------|--------|
| *E*. ViT-S | BRECQ [17] | 24.79 M | ~3157 s | 0.5395 | 0.2535 |
|  | QDrop [18] | 24.79 M | ~3678 s | 0.7166 | 0.1742 |
|  | Ours | 25.23 M | ~210 s | 0.8097 | 0.1377 |
| *E*. ViT-B | BRECQ [17] | 97.4 M | ~4069 s | 0.4692 | 0.2886 |
|  | QDrop [18] | 97.4 M | ~5820 s | 0.5785 | 0.2334 |
|  | Ours | 99.24 M | ~411 s | 0.6845 | 0.1875 |

Table 5: Performance of QSCA with varying subset sizes on NYUv2 using Depth Anything v1(W4A4).

| Subset for training | *E*. ViT-S | | |
|---------------------|------|------|--------|
|  | Time | $\delta_1 \uparrow$ | AbsRel $\downarrow$ |
| 1% | ~45 s | 0.7778 | 0.1506 |
| 3% | ~130 s | 0.8050 | 0.1395 |
| 5% | ~210 s | 0.8097 | 0.1377 |
| 10% | ~420 s | 0.8108 | 0.1375 |
| 100% | ~4200 s | 0.8177 | 0.1351 |

QDrop by a clear margin. In the 4/6 configuration, the proposed method further improves the $\delta_1$ to 0.9333 and 0.9441 while maintaining AbsRel below 0.1. Similarly, on the KITTI dataset, QSCA achieves a $\delta_1$ of 0.7273 using the ViT-S backbone under the 4/4 configuration and 0.8857 under the 4/6 configuration, significantly outperforming other methods. On more challenging datasets such as Sintel [45], ETH3D [46], and DIODE [47], as shown in Table 3, the proposed framework demonstrates strong generalization capabilities. Despite the increased scene complexity and domain shift, QSCA consistently achieves higher $\delta_1$ scores and lower AbsRel values than existing PTQ methods, effectively mitigating the detrimental effects of 4-bit quantization.

**Qualitative results.** To confirm the effectiveness of our proposed method, we visualize the predicted depth maps in Figure 3 under the W4A6 quantization setting using the ViT-S backbone on the NYUv2 [42] and KITTI [43] datasets. Compared to strong baselines such as BRECQ [17] and QDrop [18], our QSCA framework exhibits superior visual quality, preserving structural consistency and fine-grained depth transitions. On NYUv2, our method produces noticeably sharper boundaries around objects like table, and television, accurately distinguishing foreground and background with minimal artifacts. On KITTI, QSCA effectively captures linear structures such as road boundaries and fences, which are often blurred or washed out in baseline methods. These improvements are attributed to the combination of strategically inserted SCA modules and self-supervised block-wise distillation, which jointly recover spatial fidelity lost due to low-bit quantization. Notably, our results are visually closest to the full-precision FP32 outputs, demonstrating the robustness and perceptual quality of QSCA across diverse scene types.

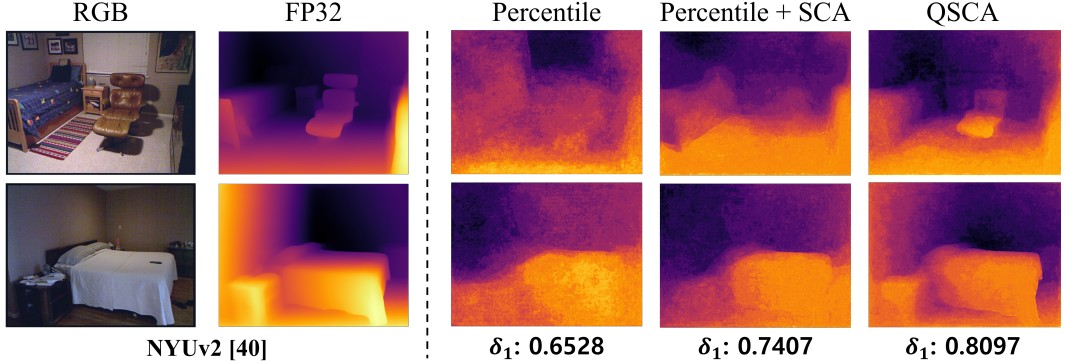

| RGB | FP32 | Percentile | Percentile + SCA | QSCA |
|-----|------|------------|------------------|------|

**NYUv2 [40]** | $\delta_1$: **0.6528** | $\delta_1$: **0.7407** | $\delta_1$: **0.8097**

Figure 4: Qualitative results on the NYUv2 dataset showing the impact of SCA modules and self-supervised distillation under 4-bit quantization.

## 4.3 Ablation study

**Efficiency and model complexity.** As shown in Table 4, we compare QSCA with existing reconstruction PTQ methods, BRECQ [17] and QDrop [18], in terms of parameter count, reconstruction time, and accuracy on the NYUv2 dataset using Depth Anything v1 with the ViT-small and ViT-base backbone. QSCA introduces a small increase of approximately 0.44 million parameters due to the insertion of lightweight SCA modules. Despite this addition, it reconstructs the quantized model in less than 210 seconds, which is significantly faster than BRECQ, whose reconstruction time exceeds 3100 seconds, and QDrop, which requires more than 3600 seconds. More importantly, it achieves substantial accuracy gains. For example, QSCA improves the $\delta_1$ score from 0.5395 in BRECQ and 0.7166 in QDrop to 0.8097, while reducing AbsRel from 0.2535 and 0.1742 respectively to 0.1377. The same trend is observed in the results for Depth Anything v2. These findings highlight that QSCA provides both faster reconstruction and better accuracy, with only a minimal increase in model size. Furthermore, Table 5 investigates the impact of training subset size on QSCA performance. As the proportion of training data increases from 1% to 100%, both $\delta_1$ and AbsRel metrics steadily improve, indicating that our method can effectively leverage even a small fraction of the training set to achieve competitive results. Notably, QSCA maintains robust performance with as little as 5% of the training data, demonstrating its efficiency and practicality in resource-constrained scenarios

**Effect of SCA Modules.** Figure 4 provides a qualitative comparison across different ablation settings. We begin with Percentile-based PTQ [44], which serves as the foundation for our approach. When inserting SCA modules without supervision (Percentile + SCA), the quality of depth prediction improves marginally, suggesting that architectural compensation alone is insufficient. The $\delta_1$ metric increases from 0.6528 to 0.7407, indicating partial recovery of structural cues. However, the full QSCA framework that integrates SCA modules with self-supervised block-wise distillation leads to clear improvements in both structural consistency and semantic preservation, with $\delta_1$ further improving to 0.8097. This progression demonstrates the effectiveness of our distillation strategy in maximizing the potential of SCA modules under 4-bit quantization.

## 5 Conclusion

In this paper, we propose QSCA, a novel framework for effective 4-bit quantization of Depth Anything. To restore performance degradation caused by quantization, we insert SCA modules into transformer and decoder blocks. These modules are trained in a self-supervised manner using only unlabeled calibration data, enabling the model to maintain high accuracy even in low-precision settings for complex depth prediction architectures. Notably, QSCA achieves competitive performance across various benchmarks while quantizing all layers to 4 bits, demonstrating its effectiveness as a practical alternative that overcomes the limitations of existing PTQ methods.

**Limitations & Future Works.** We were unable to perform direct comparisons with recent reconstruction-based PTQ methods due to limited GPU resources. Moreover, these methods are

primarily optimized for classification tasks and often fail to function reliably in constrained environments. This limitation becomes especially pronounced in dense prediction tasks such as monocular depth estimation, where memory consumption is significantly higher. Additionally, our proposed method is based on fake quantization, a technique that simulates quantization effects. In future work, we plan to design more lightweight and highly optimized quantization techniques that surpass existing resource-efficient methods, and to extend these approaches to a broad range of dense prediction tasks.

## Acknowledgments

This research was supported by the MSIT (Ministry of Science and ICT), Korea, under the ITRC (Information Technology Research Center) support program (IITP-2025-RS-2023-00260091) supervised by the IITP (Institute for Information & Communications Technology Planning & Evaluation) and development of an analog-digital mixed ultra-low power neuromorphic edge SoC (RS-2025-02263706) and the National Research Foundation of Korea (NRF) grant funded by the Korea government (MSIT)(RS-2025-16066849).

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

# Appendix

## A  Additional Quantitative Results

In this section, we provide additional quantitative results to further validate our proposed QSCA framework. Tables 6 and 7 present expanded comparisons on the NYUv2 [42] and KITTI [43] datasets, supplementing the main results in Tables 1 and 2. We compare our framework against recent PTQ methods, including PTQ4ViT [27], RepQ-ViT [36], ERQ [48], and QwT [31]. Furthermore, to demonstrate robustness across various bit-widths, we evaluate an additional W4A8 (4-bit weights and 8-bit activations) configuration alongside the existing W4A4 and W4A6 settings.

Table 6: Additional quantization results on the NYUv2 [42] for zero-shot relative depth estimation. W/A indicates the bit-width of weights and activations after quantization. *E.* denotes the encoder backbone used in the MDE architectures.

| Method | W/A | Depth Anything v1 [7] | | | | Depth Anything v2 [8] | | | |
| | | *E.* ViT-S | | *E.* ViT-B | | *E.* ViT-S | | *E.* ViT-B | |
| | | $\delta_1 \uparrow$ | AbsRel $\downarrow$ | $\delta_1 \uparrow$ | AbsRel $\downarrow$ | $\delta_1 \uparrow$ | AbsRel $\downarrow$ | $\delta_1 \uparrow$ | AbsRel $\downarrow$ |
|---|---|---|---|---|---|---|---|---|---|
| FP | 32/32 | 0.9720 | 0.0525 | 0.9791 | 0.0459 | 0.9736 | 0.0513 | 0.9770 | 0.0460 |
| MinMax [14] | 4/4 | 0.5024 | 0.2728 | 0.1972 | 1.5735 | 0.4873 | 0.2815 | 0.1102 | 2.3712 |
| Percentile [44] | 4/4 | 0.6542 | 0.2006 | 0.5430 | 0.2522 | 0.6675 | 0.1935 | 0.5379 | 0.2543 |
| BRECQ [17] | 4/4 | 0.5395 | 0.2535 | 0.4692 | 0.2886 | 0.5042 | 0.2714 | 0.4646 | 0.2910 |
| QDrop [18] | 4/4 | 0.7166 | 0.1742 | 0.5785 | 0.2334 | 0.7115 | 0.1773 | 0.5794 | 0.2332 |
| PTQ4ViT [27] | 4/4 | 0.5693 | 0.2393 | 0.5895 | 0.2294 | 0.5034 | 0.2735 | 0.5183 | 0.2694 |
| RepQ-ViT [36] | 4/4 | 0.6639 | 0.1959 | 0.5410 | 0.2539 | 0.6464 | 0.2016 | 0.5465 | 0.2507 |
| ERQ [48] | 4/4 | 0.7062 | 0.1785 | 0.6126 | 0.2182 | 0.7140 | 0.1751 | 0.4705 | 0.2883 |
| QwT [31] | 4/4 | 0.8007 | 0.1407 | 0.6486 | 0.2024 | 0.8050 | 0.1400 | 0.6589 | 0.1992 |
| Ours | 4/4 | 0.8097 | 0.1377 | 0.6845 | 0.1875 | 0.8151 | 0.1361 | 0.6845 | 0.1875 |
| MinMax [14] | 4/6 | 0.5632 | 0.2417 | 0.4973 | 0.2748 | 0.5324 | 0.2599 | 0.4738 | 0.2866 |
| Percentile [44] | 4/6 | 0.8837 | 0.1050 | 0.9071 | 0.0958 | 0.9196 | 0.0891 | 0.9355 | 0.0831 |
| BRECQ [17] | 4/6 | 0.5786 | 0.2337 | 0.5542 | 0.2462 | 0.5269 | 0.2611 | 0.5084 | 0.2697 |
| QDrop [18] | 4/6 | 0.6369 | 0.2071 | 0.6987 | 0.1823 | 0.7285 | 0.1700 | 0.7619 | 0.1560 |
| PTQ4ViT [27] | 4/6 | 0.6187 | 0.2166 | 0.6122 | 0.2201 | 0.5159 | 0.2642 | 0.6095 | 0.2275 |
| RepQ-ViT [36] | 4/6 | 0.8256 | 0.1304 | 0.8867 | 0.1058 | 0.8657 | 0.1131 | 0.9158 | 0.0918 |
| ERQ [48] | 4/6 | 0.9075 | 0.0945 | 0.9390 | 0.0766 | 0.9245 | 0.0874 | 0.9152 | 0.0914 |
| QwT [31] | 4/6 | 0.9189 | 0.0884 | 0.9089 | 0.0928 | 0.9323 | 0.0833 | 0.8944 | 0.1000 |
| Ours | 4/6 | 0.9333 | 0.0810 | 0.9441 | 0.0739 | 0.9450 | 0.0757 | 0.9468 | 0.0726 |
| MinMax [14] | 4/8 | 0.7740 | 0.1522 | 0.5934 | 0.2253 | 0.8187 | 0.1345 | 0.5465 | 0.2507 |
| Percentile [44] | 4/8 | 0.9233 | 0.0861 | 0.9588 | 0.0648 | 0.9375 | 0.0798 | 0.9621 | 0.6037 |
| BRECQ [17] | 4/8 | 0.8533 | 0.1182 | 0.6166 | 0.2152 | 0.8927 | 0.0999 | 0.6095 | 0.2193 |
| QDrop [18] | 4/8 | 0.9504 | 0.0700 | 0.9627 | 0.0599 | 0.9592 | 0.0651 | 0.9621 | 0.6010 |
| PTQ4ViT [27] | 4/8 | 0.6544 | 0.2001 | 0.6837 | 0.1877 | 0.5246 | 0.2596 | 0.6328 | 0.2190 |
| RepQ-ViT [36] | 4/8 | 0.8833 | 0.1066 | 0.9259 | 0.0864 | 0.8792 | 0.1066 | 0.9367 | 0.0788 |
| ERQ [48] | 4/8 | 0.9333 | 0.0812 | 0.9632 | 0.0608 | 0.9355 | 0.0809 | 0.9641 | 0.0597 |
| QwT [31] | 4/8 | 0.9393 | 0.0769 | 0.9348 | 0.0789 | 0.9564 | 0.0663 | 0.9300 | 0.0808 |
| Ours | 4/8 | 0.9551 | 0.0682 | 0.9542 | 0.0657 | 0.9549 | 0.0679 | 0.9635 | 0.0599 |

Table 7: Additional quantization results on the KITTI [43] for zero-shot relative depth estimation. W/A indicates the bit-width of weights and activations after quantization. *E.* denotes the encoder backbone used in the MDE architectures.

| Method | W/A | Depth Anything v1 [7] | | | | Depth Anything v2 [8] | | | |
| | | *E.* ViT-S | | *E.* ViT-B | | *E.* ViT-S | | *E.* ViT-B | |
| | | $\delta_1 \uparrow$ | AbsRel $\downarrow$ | $\delta_1 \uparrow$ | AbsRel $\downarrow$ | $\delta_1 \uparrow$ | AbsRel $\downarrow$ | $\delta_1 \uparrow$ | AbsRel $\downarrow$ |
|---|---|---|---|---|---|---|---|---|---|
| FP | 32/32 | 0.9369 | 0.0818 | 0.9396 | 0.0804 | 0.9340 | 0.0832 | 0.9389 | 0.0814 |
| MinMax [14] | 4/4 | 0.3441 | 0.3770 | 0.2058 | 1.9612 | 0.3423 | 0.3938 | 0.0832 | 4.4358 |
| Percentile [44] | 4/4 | 0.4099 | 0.3418 | 0.3327 | 0.3876 | 0.3780 | 0.3668 | 0.3275 | 0.3932 |
| BRECQ [17] | 4/4 | 0.3522 | 0.3719 | 0.3160 | 0.3989 | 0.3344 | 0.3906 | 0.3175 | 0.3990 |
| QDrop [18] | 4/4 | 0.3234 | 0.3934 | 0.3338 | 0.3855 | 0.3748 | 0.3620 | 0.4082 | 0.3412 |
| PTQ4ViT [27] | 4/4 | 0.4106 | 0.3439 | 0.3251 | 0.3923 | 0.4187 | 0.3308 | 0.3200 | 0.3972 |
| RepQ-ViT [36] | 4/4 | 0.4159 | 0.3434 | 0.5410 | 0.2539 | 0.6464 | 0.2016 | 0.5465 | 0.2507 |
| ERQ [48] | 4/4 | 0.4847 | 0.3178 | 0.4241 | 0.3528 | 0.4616 | 0.3176 | 0.4066 | 0.3490 |
| QwT [31] | 4/4 | 0.6862 | 0.1802 | 0.5867 | 0.2417 | 0.6941 | 0.1951 | 0.5346 | 0.2539 |
| Ours | 4/4 | 0.7273 | 0.1874 | 0.6203 | 0.2365 | 0.6794 | 0.2067 | 0.6174 | 0.2296 |
| MinMax [14] | 4/6 | 0.4467 | 0.3290 | 0.3586 | 0.3769 | 0.3609 | 0.3744 | 0.3161 | 0.3982 |
| Percentile [44] | 4/6 | 0.8558 | 0.1269 | 0.7580 | 0.1687 | 0.8613 | 0.1283 | 0.7219 | 0.1866 |
| BRECQ [17] | 4/6 | 0.5223 | 0.2851 | 0.4035 | 0.3529 | 0.4320 | 0.3332 | 0.4093 | 0.3497 |
| QDrop [18] | 4/6 | 0.6052 | 0.2408 | 0.5897 | 0.2611 | 0.5351 | 0.2790 | 0.5268 | 0.3037 |
| PTQ4ViT [27] | 4/6 | 0.4446 | 0.3230 | 0.3488 | 0.3848 | 0.3931 | 0.3614 | 0.3308 | 0.3917 |
| RepQ-ViT [36] | 4/6 | 0.7888 | 0.1502 | 0.5610 | 0.2602 | 0.8293 | 0.1292 | 0.8109 | 0.1458 |
| ERQ [48] | 4/6 | 0.8629 | 0.1244 | 0.8705 | 0.1133 | 0.8770 | 0.1144 | 0.8685 | 0.1115 |
| QwT [31] | 4/6 | 0.8844 | 0.1152 | 0.7292 | 0.1759 | 0.8872 | 0.1169 | 0.7700 | 0.1550 |
| Ours | 4/6 | 0.8857 | 0.1161 | 0.8722 | 0.1174 | 0.8893 | 0.1124 | 0.8873 | 0.1067 |
| MinMax [14] | 4/8 | 0.4215 | 0.3486 | 0.4845 | 0.3275 | 0.4237 | 0.3529 | 0.4376 | 0.3327 |
| Percentile [44] | 4/8 | 0.8740 | 0.1111 | 0.8564 | 0.1263 | 0.8892 | 0.1051 | 0.9003 | 0.1045 |
| BRECQ [17] | 4/8 | 0.4070 | 0.3532 | 0.4513 | 0.3317 | 0.4575 | 0.3194 | 0.4493 | 0.3239 |
| QDrop [18] | 4/8 | 0.4883 | 0.3071 | 0.5422 | 0.2754 | 0.5368 | 0.2713 | 0.5616 | 0.2688 |
| PTQ4ViT [27] | 4/8 | 0.4580 | 0.3224 | 0.3550 | 0.3813 | 0.3909 | 0.3538 | 0.3205 | 0.3969 |
| RepQ-ViT [36] | 4/8 | 0.8145 | 0.1375 | 0.6436 | 0.2227 | 0.8431 | 0.1212 | 0.8683 | 0.1136 |
| ERQ [48] | 4/8 | 0.8801 | 0.1164 | 0.8937 | 0.1005 | 0.8881 | 0.1040 | 0.9043 | 0.0988 |
| QwT [31] | 4/8 | 0.8970 | 0.1044 | 0.7362 | 0.1698 | 0.9017 | 0.1061 | 0.8228 | 0.1328 |
| Ours | 4/8 | 0.9022 | 0.1019 | 0.8963 | 0.1045 | 0.9010 | 0.1020 | 0.9161 | 0.0931 |

# B    Additional Ablation Studies

In this section, we present additional ablation studies to analyze the key components and hyperparameters of our proposed method.

**Computational Cost of SCA.**    To analyze the computational cost of our proposed SCA module, we performed theoretical and empirical evaluations. We show the detailed results presented in Table 8. To evaluate the overhead of SCA modules, we calculated the model size difference with and without the SCA under 4-bit quantization. Furthermore, to approximate the latency impact, we converted an 8-bit quantized model into a TensorRT engine and measured its latency and peak memory on an RTX4090 GPU, both before and after adding the SCA modules. While this analysis is not a full replacement for actual on-device measurements at 4-bit precision, it offers a practical approximation. As a result, the inclusion of the SCA module introduces only a marginal computational overhead. This minimal cost is a reasonable trade-off considering the substantial performance gains provided by SCA.

**Ablation on Hyperparameter $\lambda_1$.**    We conducted an ablation study on the hyperparameter $\lambda_1$ to analyze its sensitivity, as shown in Table 9. The performance is optimal when $\lambda_1$ is set to 1000. This experiment validates our choice of $\lambda_1$ for the main experiments, as it demonstrates a clear trend where values in this range yield the best performance.

Table 8: Computational efficiency comparison with and without the proposed SCA.

|  | Model size (MB) | | Latency (ms) | | Peak memory (MiB) | |
|---|---|---|---|---|---|---|
|  | E. ViT-S | E. ViT-B | E. ViT-S | E. ViT-B | E. ViT-S | E. ViT-B |
| FP32 | 94.62 | 371.90 | 7.64 | 17.27 | $\sim 334.8$ | $\sim 826.6$ |
| (w/ o) SCA | 14.49 | 52.93 | 1.81 | 3.57 | $\sim 78.4$ | $\sim 229.8$ |
| (w/ ) SCA | 16.27 | 60.02 | 1.82 | 3.79 | $\sim 84.3$ | $\sim 240.4$ |

Table 9: Ablation study on the hyperparameter $\lambda_1$.

| $\lambda_1$ | 0.1 | 1 | 10 | 100 | 1000 |
|---|---|---|---|---|---|
| $\delta_1 \uparrow$ | $0.7873 \pm 0.008$ | $0.7885 \pm 0.010$ | $0.7965 \pm 0.008$ | $0.8032 \pm 0.010$ | $0.8097 \pm 0.004$ |
| AbsRel $\downarrow$ | $0.1461 \pm 0.003$ | $0.1455 \pm 0.003$ | $0.1424 \pm 0.002$ | $0.1399 \pm 0.004$ | $0.1377 \pm 0.001$ |

