# OpenReview forum: "QSCA: Quantization with Self-Compensating Auxiliary for Monocular Depth Estimation"
_NeurIPS.cc/2025/Conference — NeurIPS 2025 poster_

### Official Review · Reviewer_hdHX · 2025-07-02

**Clarity:** 3
**Significance:** 3
**Originality:** 3
**Rating:** 5
**Confidence:** 4

**Summary:**

This paper presents QSCA (quantization with self-compensating auxiliary), a post-training quantization framework for large-scale monocular depth estimation models (depth-anything-v1/v2). QSCA integrates lightweight self-compensating auxiliary modules (SCA) into transformer encoder and decoder blocks and trains them via self-supervised distillation from the full-precision teacher model, without requiring any ground truth depth labels. The authors demonstrate that inserting and fine-tuning these modules on just 5% of calibration data recovers much of the performance lost under aggressive 4-bit weight/activation quantization. Evaluations on five zero-shot benchmarks (NYUv2, KITTI, Sintel, ETH3D, DIODE) show substantial improvements in both accuracy and AbsRel error over state-of-the-art post-training quantization baselines, while adding minimal parameters and achieving fast adaptation times.

**Questions:**

1. It would be helpful if the authors could report the mean and standard deviation of δ₁ (and other key metrics) over 3–5 independent calibration splits to demonstrate the stability of the results; a low variance (e.g., std < 0.5 %) would reinforce confidence, whereas a high variance (> 1%) might indicate sensitivity to sample choice.
2. Could the authors include a direct comparison against QwT and a well-known QAT method (such as LSQ+ or DoReFa) under identical 4-bit settings on NYUv2 and KITTI? Demonstrating that QSCA matches or exceeds these baselines would underscore its advantage.
3. It would strengthen the manuscript to measure and report end-to-end inference latency, throughput (FPS), and peak memory on a representative hardware platform or GPU. Showing minimal overhead (e.g., < 5% slowdown) would validate practical deployment claims.
4. I suggest an ablation study that swaps the module types—using convolutional SCAs in the encoder and linear SCAs in the decoder and other alternative module types. And report the impact on δ₁ and AbsRel. A clear performance drop in these variants would justify the authors’ architectural choices.
5. Could the authors profile the closed-form initialization step and report how much of the 210 sec adaptation time is spent on the per-block Gram matrix inversions? In addition, did you observe any numerical instability (e.g., failed inversions or large weight magnitudes) when using only 5% of noisy calibration data?
6. How sensitive is QSCA to the regularization weight 𝜆 in Eq. (6)? Reporting performance for at least two 𝜆 values would show whether the method works robustly or requires careful tuning.

**Ethical Concerns:**

["NO or VERY MINOR ethics concerns only"]

**Final Justification:**

The rebuttal addressed all my concerns. The authors added multi-split stability analysis (std <0.5%), direct QwT comparisons showing consistent improvements, λ-sensitivity experiments, and profiling of the initialization step confirming negligible overhead. While real on-device 4-bit latency remains future work, their proxy evaluation and parameter/memory analysis are reasonable. Given the strengthened empirical evidence and sound justifications, I am increasing my score to 5 (Accept).

**Limitations:**

Yes

**Paper Formatting Concerns:**

No major formatting issues observed.

**Quality:**

3

**Strengths And Weaknesses:**

Strengths:
1. The paper introduces self-supervised SCA modules that compensate for quantization error without requiring any depth labels, enabling truly label-free post-training adaptation.
2. Demonstrates substantial empirical gains in accuracy over strong PTQ baselines under 4-bit quantization across all 5 datasets.
3. Achieves fast adaptation (210s) and minimal parameter overhead, making it practical for real-world deployment.
4. Validated on five diverse zero-shot benchmarks (NYUv2, KITTI, Sintel, ETH3D, DIODE), showing robust generalization under aggressive quantization.
5. The method is highly modular and architecture-agnostic, allowing seamless integration of these SCA modules into a wide range of transformer-based models without altering the original network weights.
6. The closed-form ridge-regression initialization for each SCA (Eq. 6) provides a principled starting point that immediately corrects large quantization errors, rather than relying on random or heuristic initialization.
7. Block-wise self-supervised distillation (feature-level + SILog loss, Eqs. 8–9) propagates rich correction signals into both intermediate representations and final outputs, which is distinctive in the PTQ setting.
8. The split design of linear projections in encoder blocks vs. small convolutional layers in decoder blocks is well motivated by feature dimensionality and spatial structure, keeping modules lightweight yet effective.

Weakness:
1. Table 5 reports accuracies for different training subsets, but it lacks error bars or results over multiple random splits, so it’s unclear whether the gains at, for example, 5% calibration are stable across different sample choices.
2. Although the approach is inspired by QwT, the paper does not include a direct empirical comparison, making it difficult to quantify the advantages of the proposed self-supervised distillation.
3. Reconstruction time is provided, but end-to-end inference latency, throughput, and memory usage on representative hardware are not measured, limiting assessment of real-world deployment feasibility.
4. While the use of linear projections in encoder SCAs and small convolutions in decoder SCAs is well motivated by feature dimensionality and spatial structure, the authors do not empirically compare these against alternative lightweight module designs (e.g. small MLPs or depthwise convolutions) to confirm that this split yields the best trade-off.
5. For better readability, please fix the Figure 2(b) caption and figure label can be improved by adding the x- and y-axes and chart title.
6. Distillation is confined to the SCA-augmented blocks. Exploring a more global distillation objective (e.g., attention-map matching) might yield additional gains but is left unexamined.
7. Initializing each SCA via Eq.(6) involves inverting a $d_{\mathrm{in}}\times d_{\mathrm{in}}$ matrix (cost $O(d_{\mathrm{in}}^3)$). Please clarify if this overhead is noticeable for hidden sizes and whether calibration noise affects numerical stability.

---

> ### Author Rebuttal · Authors · 2025-07-30
>
> >**Q1:** It would be helpful if the authors could report the mean and standard deviation of δ₁(and other key metrics) over 3–5 independent calibration splits to demonstrate the stability of the results; a low variance(e.g., std<0.5%) would reinforce confidence, whereas a high variance(>1%) might indicate sensitivity to sample choice.
>
> **A1:** To evaluate the stability of our method under different calibration data selections, we conducted an additional experiment using five independent runs with different seeds for calibration and training. For each run, we randomly sampled a different 5% subset of the training data and measured the performance of QSCA. We then computed the mean and standard deviation of the δ₁ and AbsRel. The results demonstrate that our method consistently achieves stable performance, with standard deviations mostly remaining below 0.5%.
> We report the average performance and std for each dataset in the following table:
> |Dataset|NYU||||KITTI||||
> -|-|-|-|-|-|-|-|-
> ||*E.* ViT-S||*E.* ViT-B||*E.* ViT-S||*E.* ViT-B
> W/A|δ₁↑|AbsRel↓|δ₁↑|AbsRel↓|δ₁↑|AbsRel↓|δ₁↑|AbsRel↓
> 4/4|0.8097±0.004|0.1377±0.001|0.6252±0.014|0.2141±0.006|0.7273±0.005|0.1874±0.003|0.6004±0.028|0.2523±0.013
> 4/6|0.9160±0.005|0.0902±0.003|0.9438±0.007|0.0743±0.004|0.8857±0.009|0.1161±0.006|0.8722±0.013|0.1174±0.006
> >**Q2**: Could the authors include a direct comparison against QwT and a well-known QAT method(such as LSQ+ or DoReFa) under identical 4-bit settings on NYUv2 and KITTI? Demonstrating that QSCA matches or exceeds these baselines would underscore its advantage.
>
> **A2**: To reflect the reviewer’s comment, we included QwT in our experimental comparisons.
> Our work specifically targets MDE foundation models like Depth Anything, which are pretrained to predict relative depth using massive datasets. Applying QAT(such as LSQ+ or DoReFa) to such models would require full retraining with access to the original dataset, which is not feasible under realistic hardware constraints. QSCA demonstrates consistently better performance and robustness in most cases under the same 4-bit settings on NYUv2 and KITTI. We would also like to highlight the parameter efficiency of our method. Compared to QwT, our QSCA framework results in a much smaller increase in the number of parameters. This makes our approach more lightweight and practical for deployment in resource-constrained environments.
> Dataset||NYU||||KITTI||||
> -|-|-|-|-|-|-|-|-|-
> |||*E.* ViT-S||*E.* ViT-B||*E.* ViT-S||*E.* ViT-B
> Method|W/A|δ₁↑|AbsRel↓|δ₁↑|AbsRel↓|δ₁↑|AbsRel↓|δ₁↑|AbsRel↓
> QwT|4/4|0.8007|0.1407|0.6486|0.2024|0.6862|0.1802|0.5867|0.2417
> Ours|4/4|0.8097±0.004|0.1377±0.001|0.6252±0.014|0.2141±0.006|0.7273±0.005|0.1874±0.003|0.6004±0.028|0.2523±0.013
> QwT|4/6|0.9049|0.0956|0.9089|0.0928|0.8844|0.1152|0.7292|0.1759
> Ours|4/6|0.9160±0.005|0.0902±0.003|0.9438±0.007|0.0743±0.004|0.8857±0.009|0.1161±0.006|0.8722±0.013|0.1174±0.006
>
> |Method|*E.* ViT-S|*E.* ViT-B
> -|-|-
> FP32|24.79M|97.4M
> QwT|26.56M|104.56M
> Ours|25.23M|99.24M
> >**Q3:** It would strengthen the manuscript to measure and report end-to-end inference latency, throughput(FPS), and peak memory on a representative hardware platform or GPU. Showing minimal overhead(e.g.,<5% slowdown) would validate practical deployment claims.
>
> **A3:** We focused on validating the effectiveness of the QSCA framework using fake quantization, which is widely adopted for simulating low-precision behavior without generating actual quantized models. Fake quantization is widely used, with the purpose of simulation the performance changes depending on the conducted quantization levels.  While this approach provides reliable insights into quantization-aware performance, it does not enable direct measurement of latency or memory usage on real devices. We acknowledge this as a limitation of the current study, and plan to incorporate comprehensive on-device evaluations in future work to better assess the practical deployment aspects of our framework.
> To address this limitation, we additionally performed a both theoretical and empirical analysis to estimate the deployability of our framework. We calculated the model size difference with and without the SCA modules under 4-bit quantization to evaluate parameter efficiency. Furthermore, we converted the 8-bit quantized model into a TensorRT engine and measured its latency on an RTX4090 GPU. We then added the SCA modules and re-measured latency to analyze the overhead introduced by QSCA. Although these analyses are not full replacements for actual on-device measurements at 4-bit precision, they offer a practical approximation.
>
> ||Model size(MB)||Latency(ms)||Peak memory(MiB)||
> -|-|-|-|-|-|-
> ||*E.* ViT-S|*E.* ViT-B|*E.* ViT-S|*E.* ViT-B|*E.* ViT-S|*E.* ViT-B
> FP32|94.62|371.90|7.64|17.27|~334.8|~826.6
> (w/o)SCA|14.49|52.93|1.81|3.57|~78.4|~229.8
> (w/)SCA|16.27|60.02|1.82|3.79|~84.3|~240.4
> >**Q4:** I suggest an ablation study that swaps the module types—using convolutional SCAs in the encoder and linear SCAs in the decoder and other alternative module types. And report the impact on δ₁ and AbsRel. A clear performance drop in these variants would justify the authors’ architectural choices.
>
> **A4:** We have also given careful consideration to the proposed combination of a ViT encoder with a convolutional (conv) SCA and a CNN decoder with a linear SCA.
> Our analysis is as follows:
> For an input tensor $x$ with the shape $[B,L,D]$, the ViT encoder produces an output tensor $y$ with the same shape, $[B,L,D]$. Therefore, according to Eq. (5), the weight is initialized with a $[D, D]$ shape.
> To apply a conv SCA to the ViT encoder, the input tensor must be reshaped to $[B,C_{\text{in}},H_{\text{in}},W_{\text{in}}]$, and the weights must be reshaped to $[C_{\text{out}},C_{\text{in}},K,K]$.
> For example, if we set $H_{\text{in}} \times W_{\text{in}}=L$ and $C_{\text{in}}=D$, the total number of weight elements is $D^2=C_{\text{out}}\times C_{\text{in}}\times K^2$, which implies $C_{\text{out}}=D/K^2$. The output tensor from this conv SCA, after adjusting stride and padding to ensure $H_{\text{out}}=H_{\text{in}}$ and $W_{\text{out}} = W_{\text{in}}$, will have the shape $[B,D/K^2,H_{\text{in}},W_{\text{in}}]$. Consequently, the conv SCA produces an output with a different number of channels. Considering that the ViT encoder must maintain the same input and output shape, an additional conv layer would be necessary to restore the channel dimension of the conv SCA's output from $D/K^2$ back to $D$.
> In other words, connecting a conv SCA to the ViT encoder cannot be achieved with a single conv layer, which contradicts our intended "simplest module design".
> Similarly, if we were to set $C_{\text{out}}=D$, it would result in $C_{\text{in}}=D/K^2$. This, in turn, would require an additional preceding conv layer to reduce the input tensor's channel dimension from $D$ to $D/K^2$.
> Connecting an additional layer makes it impossible to compute our proposed QSCA weight initialization in a closed-form solution. If the weight initialization cannot be performed, more epochs would be required in the self-supervised learning step to improve performance.
> Based on this analysis, we conclude that our proposed combination of a ViT encoder with a linear SCA and a CNN decoder with a conv SCA is the optimal architecture.
> >**Q5.** Could the authors profile the closed-form initialization step and report how much of the 210sec adaptation time is spent on the per-block Gram matrix inversions? In addition, did you observe any numerical instability(e.g., failed inversions or large weight magnitudes) when using only 5% of noisy calibration data?
>
> **A5.1:** We profiled the closed-form initialization step to measure the time spent on the Gram matrix inversion for each block. The results show that this step is computationally efficient, requiring approximately 1.45ms per block for E.ViT-S and 3.5ms per block for E.ViT-B, which is negligible relative to the total adaptation time of 210s.
>
> **A.5.2:** Regarding the second part of the question about numerical instability when using 5% of noisy calibration data, we would like to clarify that the closed-form initialization step in our framework does not rely on any 5% subset of train dataset. The dataset is used exclusively during the subsequent self-supervised learning phase. Therefore, this part of the question overlaps with Q1, where we already demonstrated the statistical robustness of our method under varying calibration subsets. As shown in table in Q1, the standard deviations of δ₁ and AbsRel metrics remained consistently low, suggesting that our method is robust to sample variations and does not exhibit numerical instability.
> >**Q6:** How sensitive is QSCA to the regularization weight λ in Eq.(6)? Reporting performance for at least two λ values would show whether the method works robustly or requires careful tuning.
>
> **A6:** We conducted additional ablation studies on the NYUv2 dataset using the E.ViT-S backbone with 4-bit quantization, where λ was varied across a wide range {0, 0.01, 0.1, 1, 10, 100, 1000}.
>
> The results shown in the table below indicate that while QSCA maintains a relatively stable performance for moderate values of λ such as 1 to 100, the best results are achieved when λ is appropriately tuned. For instance, performance gradually improves as λ increases, with the highest accuracy obtained at λ equals 1000, which we used as the default in our main experiments.
>
> This suggests that λ plays an important role in balancing the reconstruction and regularization terms, and some degree of tuning is indeed beneficial to fully leverage the proposed framework. We will clarify this point in the revised manuscript to guide future users in practical settings.
> λ|0.01|0.1|0|1|10|100|1000
> -|-|-|-|-|-|-|-
> δ₁↑|0.7877±0.009|0.7873±0.008|0.7864±0.009|0.7885±0.010|0.7965±0.008|0.8032±0.010|0.8097±0.004
> AbsRel↓|0.1457±0.003|0.1461±0.003|0.1466±0.003|0.1455±0.003|0.1424±0.002|0.1399±0.004|0.1377±0.001

---

> > ### Comment · Reviewer_hdHX · 2025-08-07
> >
> > Thank you for the rebuttal. All of my concerns have been addressed.

---

> > > ### Author Response · Authors · 2025-08-08
> > >
> > > Dear Reviewer hdHX,
> > >
> > > Thank you for acknowledging our efforts in addressing your concerns. We appreciate your constructive feedback, which was not only valuable for improving the manuscript but also for giving us the opportunity to reflect more deeply on our research.
> > >
> > > Sincerely,
> > >
> > > Authors

---

### Official Review · Reviewer_j8e1 · 2025-07-03

**Clarity:** 3
**Significance:** 3
**Originality:** 3
**Rating:** 5
**Confidence:** 3

**Summary:**

To reduce the computational cost of current depth foundation models, this paper proposes a 4-bit post-training quantization method, which successfully apply 4-bit quantization across all layers. The experimental results demonstrate superior performance over existing works.

**Questions:**

- To better demonstrate the practical advantages of the proposed method, it would be helpful to compare its inference time and GPU memory consumption against the full-precision baseline.

**Ethical Concerns:**

["NO or VERY MINOR ethics concerns only"]

**Final Justification:**

Thanks for the detailed response. My concerns are addressed and thus I decide to raise my rating to 5.

**Limitations:**

The limitation is discussed in the main text.

**Paper Formatting Concerns:**

There is no formatting concern.

**Quality:**

3

**Strengths And Weaknesses:**

Strength
- The proposed method have practice value.
- The proposed method demonstrates superior performance over existing works.
- The paper is well written and easy to follow.

Weakness
- The compared methods listed in Tables 1, 2, and 3 are outdated; the most recent baseline, QDrop, dates back to 2022. Including more recent works would strengthen the evaluation.
- The reported results in Table 1, 2 and 3 focus predominantly on 4-bit quantization. It is recommended to include evaluations under other bit widths—such as 8-bit—to enable a more comprehensive comparison and better understand the method’s flexibility and performance trade-offs.

---

> ### Author Rebuttal · Authors · 2025-07-30
>
> >**Q1:**  To better demonstrate the practical advantages of the proposed method, it would be helpful to compare its inference time and GPU memory consumption against the full-precision baseline.
>
> **A1:** In this study, we focused on validating the effectiveness of the QSCA framework using fake quantization, which is widely adopted for simulating low-precision behavior without generating actual quantized models. Fake quantization is widely used, with the purpose of simulation the performance changes depending on the conducted quantization levels.  While this approach provides reliable insights into quantization-aware performance, it does not enable direct measurement of latency or memory usage on real devices. We acknowledge this as a limitation of the current study, and plan to incorporate comprehensive on-device evaluations in future work to better assess the practical deployment aspects of our framework.
> To address this limitation, we additionally performed a both theoretical and empirical analysis to estimate the deployability of our framework. We calculated the model size difference with and without the SCA modules under 4-bit quantization to evaluate parameter efficiency. Furthermore, we converted the 8-bit quantized model into a TensorRT engine and measured its latency on an RTX4090 GPU. We then added the SCA modules and re-measured latency to analyze the overhead introduced by QSCA. Although these analyses are not full replacements for actual on-device measurements at 4-bit precision, they offer a practical approximation.
> As shown in the table below, our method exhibits minimal overhead in memory consumption compared to the baseline (w/o SCA). Specifically, the peak memory usage increased by only ~5.9 MiB for ViT-S and ~10.6 MiB for ViT-B.
>
> ||Model size(MB)||Latency(ms)||Peak memory(MiB)||
> -|-|-|-|-|-|-
> |Model|*E.* ViT-S|*E.* ViT-B|*E.* ViT-S|*E.* ViT-B|*E.* ViT-S|*E.* ViT-B
> FP32|94.62|371.90|7.64|17.27|~334.8|~826.6
> (w/o)SCA|14.49|52.93|1.81|3.57|~78.4|~229.8
> (w/)SCA|16.27|60.02|1.82|3.79|~84.3|~240.4
>
> >**Q2:** The compared methods listed in Tables 1, 2, and 3 are outdated; the most recent baseline, QDrop, dates back to 2022. Including more recent works would strengthen the evaluation.
>
> **A2:**  We agree with the reviewer, that the compared baselines may not be recent enough for a fair comparison and therefore our results might not be validating the strengths of our method. To address the concern of the reviewer regarding this missing comparison, we have included RepQ-ViT and ERQ as recent optimization-based PTQ baselines, and QwT as a hybrid PTQ/QAT method. Despite being alternatives, QSCA consistently outperforms them in accuracy and robustness, especially under low-bit settings, while the performance remains stable across different evaluation datasets.
>
> |  | Dataset | NYU |  |  |  | KITTI |  |  |  |
> | --- | --- | --- | --- | --- | --- | --- | --- | --- | --- |
> | Model  | | *E.* ViT-S |  | *E.* ViT-B |  | *E.* ViT-S |  | *E.* ViT-B |  |
> | Method | W/A | δ₁↑|AbsRel↓ | δ₁↑|AbsRel↓ | δ₁↑|AbsRel↓l | δ₁↑|AbsRel↓ |
> | MinMax | 4/4 | 0.5024 | 0.2728 | 0.1972 | 1.5735 | 0.3441 | 0.3770 | 0.2058 | 1.9612 |
> | Percentile | 4/4 | 0.6542 | 0.2006 | 0.5430 | 0.2522 | 0.4099 | 0.3418 | 0.3327 | 0.3876 |
> | BRECQ | 4/4 | 0.5395 | 0.2535 | 0.4692 | 0.2886 | 0.3522 | 0.3418 | 0.3160 | 0.3989 |
> | Qdrop | 4/4 | 0.7166 | 0.1742 | 0.5785 | 0.2334 | 0.3234 | 0.3719 | 0.3338 | 0.3855 |
> | PTQ4ViT | 4/4 | 0.5693 | 0.2393 | 0.5895 | 0.2294 | 0.4106 | 0.3439 | 0.3251 | 0.3923 |
> | RepQ-ViT | 4/4 | 0.6639 | 0.1959 | 0.5410 | 0.2539 | 0.4159 | 0.3434 | 0.3164 | 0.3982 |
> | ERQ | 4/4 | 0.7062 | 0.1785 | 0.6126 | 0.2182 | 0.4847 | 0.3178 | 0.4241 | 0.3528 |
> | QwT | 4/4 | 0.8007 | 0.1407 | 0.6486 | 0.2024 | 0.6862 | 0.1802 | 0.5867 | 0.2417 |
> | Ours | 4/4 | 0.8097±0.004 | 0.1377±0.001 | 0.6252±0.014 | 0.2141±0.006 | 0.7273±0.005 | 0.1874±0.003 | 0.6004±0.028 | 0.2523±0.013 |
> |||||||||||
> | MinMax | 4/6 | 0.5632 | 0.2417 | 0.4973 | 0.2748 | 0.4467 | 0.3290 | 0.3586 | 0.3769 |
> | Percentile | 4/6 | 0.8837 | 0.1050 | 0.9071 | 0.0958 | 0.8558 | 0.1269 | 0.7580 | 0.1687 |
> | BRECQ | 4/6 | 0.5786 | 0.2337 | 0.5542 | 0.2462 | 0.5223 | 0.2851 | 0.4035 | 0.3529 |
> | Qdrop | 4/6 | 0.6369 | 0.2071 | 0.6987 | 0.1823 | 0.6052 | 0.2408 | 0.5897 | 0.2611 |
> | PTQ4ViT | 4/6 | 0.6187 | 0.2166 | 0.6122 | 0.2201 | 0.4446 | 0.3230 | 0.3488 | 0.3848 |
> | RepQ-ViT | 4/6 | 0.8256 | 0.1304 | 0.8867 | 0.1058 | 0.7888 | 0.1502 | 0.5610 | 0.2602 |
> | ERQ | 4/6 | 0.9075 | 0.0945 | 0.9390 | 0.0766 | 0.8629 | 0.1244 | 0.8705 | 0.1133 |
> | QwT | 4/6 | 0.9049 | 0.0956 | 0.9089 | 0.0928 | 0.8844 | 0.1152 | 0.7292 | 0.1759 |
> | Ours | 4/6 | 0.9160±0.005 | 0.0902±0.003 | 0.9438±0.007 | 0.0743±0.004 | 0.8857±0.009 | 0.1161±0.006 | 0.8722±0.013 | 0.1174±0.006 |
>
> >**Q3:** The reported results in Table 1, 2 and 3 focus predominantly on 4-bit quantization. It is recommended to include evaluations under other bit widths—such as 8-bit—to enable a more comprehensive comparison and better understand the method’s flexibility and performance trade-offs.
>
> **A3:** Our primary focus in this work is on 4-bit quantization, as it represents a challenging yet practically valuable setting for deploying foundation models for monocular depth estimation on resource-constrained hardware. We sincerely appreciate the reviewer’s suggestion to expand our evaluation to include a wider range of bit-widths.
> Therefore, to demonstrate the flexibility and broader applicability of our method, we additionally included 8-bit quantization results, shown in the table below. The results show that QSCA performs consistently well even at higher bit-widths, further validating its generalizability and robustness.
>
> |  | Dataset | NYU |  |  |  | KITTI |  |  |  |
> | --- | --- | --- | --- | --- | --- | --- | --- | --- | --- |
> | Model  | | *E.* ViT-S |  | *E.* ViT-B |  | *E.* ViT-S |  | *E.* ViT-B |  |
> | Method | W/A |δ₁↑|AbsRel↓ | δ₁↑|AbsRel↓ | δ₁↑|AbsRel↓ |δ₁↑|AbsRel↓ |
> | MinMax | 4/8 | 0.7740 | 0.1522 | 0.5934 | 0.2253 | 0.4215 | 0.3486 | 0.4845 | 0.3275 |
> | Percentile | 4/8 | 0.9233 | 0.0861 | 0.9588 | 0.0648 | 0.8740 | 0.1111 | 0.8564 | 0.1263 |
> | BRECQ | 4/8 | 0.8533 | 0.1182 | 0.6166 | 0.2152 | 0.4070 | 0.3532 | 0.4513 | 0.3317 |
> | QDrop | 4/8 | 0.9504 | 0.0700 | 0.9627 | 0.0599 | 0.4883 | 0.3071 | 0.5422 | 0.2754 |
> | PTQ4ViT | 4/8 | 0.6544 | 0.2001 | 0.6837 | 0.1877 | 0.4580 | 0.3224 | 0.3550 | 0.3813 |
> | RepQ-ViT | 4/8 | 0.8833 | 0.1066 | 0.9259 | 0.0864 | 0.8145 | 0.1375 | 0.6436 | 0.2227 |
> | ERQ | 4/8 | 0.9333 | 0.0812 | 0.9632 | 0.0608 | 0.8801 | 0.1164 | 0.8937 | 0.1005 |
> | QwT | 4/8 | 0.9393 | 0.0769 | 0.9348 | 0.0789 | 0.8970 | 0.1044 | 0.7362 | 0.1698 |
> | Ours | 4/8 | 0.9551±0.001 | 0.0682±0.001 | 0.9542±0.006 | 0.0657±0.006 | 0.9022±0.003 | 0.1019±0.004 | 0.8963±0.010 | 0.1045±0.005 |
> |||||||||||
> | MinMax | 8/8 | 0.8714 | 0.1115 | 0.6028 | 0.2224 | 0.4328 | 0.3313 | 0.4092 | 0.3501 |
> | Percentile | 8/8 | 0.9662 | 0.0591 | 0.9743 | 0.0493 | 0.9288 | 0.0893 | 0.9317 | 0.0900 |
> | BRECQ | 8/8 | 0.8720 | 0.1113 | 0.6313 | 0.2101 | 0.4342 | 0.3111 | 0.4371 | 0.3367 |
> | QDrop | 8/8 | 0.9164 | 0.0893 | 0.6730 | 0.1912 | 0.4642 | 0.3083 | 0.4990 | 0.3056 |
> | PTQ4ViT | 8/8 | 0.9439 | 0.0829 | 0.9220 | 0.0922 | 0.8393 | 0.1374 | 0.8476 | 0.1346 |
> | RepQ-ViT | 8/8 | 0.9668 | 0.0590 | 0.9743 | 0.0493 | 0.9298 | 0.0885 | 0.9320 | 0.0896 |
> | ERQ | 8/8 | 0.9658 | 0.0594 | 0.9751 | 0.0498 | 0.9317 | 0.0870 | 0.9375 | 0.0818 |
> | QwT | 8/8 | 0.9672 | 0.0569 | 0.9680 | 0.0574 | 0.9319 | 0.0871 | 0.8227 | 0.1312 |
> | Ours | 8/8 | 0.9716±0.001 | 0.0534±0.001 | 0.9736±0.006 | 0.0507±0.006 | 0.9336±0.000 | 0.0846±0.000 | 0.9175±0.004 | 0.0946±0.003 |

---

### Official Review · Reviewer_89H7 · 2025-07-04

**Clarity:** 3
**Significance:** 3
**Originality:** 3
**Rating:** 4
**Confidence:** 2

**Summary:**

The paper proposes a quantization method for monocular depth estimation technique. The proposed Quantization with Self-Compensating Auxiliary for Monocular Depth Estimation (QSCA), is 4-bit post training quantization. The QSCA is ensemble of proposed Self-Compensating Auxiliary (SCA) with existing encoder and decoder blocks, this enables the model to recover the lost performance during the quantization.

**Questions:**

please refer weaknesses for questions about proposed QSCA

**Ethical Concerns:**

["NO or VERY MINOR ethics concerns only"]

**Final Justification:**

Thankyou Authors for providing rebuttal. After reading the reviews from other reviewers and authors responses I would like to maintain my initial rating.

**Limitations:**

limitations were discussed adequately in the main paper

**Paper Formatting Concerns:**

no concerns

**Quality:**

3

**Strengths And Weaknesses:**

Strengths:
- One of the first method to propose 4-bit quantization technique for the monocular depth estimation.
- Authors compute a residue that improves the lost quality of feature map at each layer of the transformer block, this is implemented residual manner where SCA takes the input to the quantized block and adding the auxiliary output to the quantized output of the blocks.
- the authors use self-supervised learning strategy using the full-precision model F as the teacher, and train the quantized model
- the authors conducted extensive experiments using Depth Anything v1 and Depth Anything v2
- significant improvements in the inference times

Weaknesses:
- Can the proposed QSCA method work on video depth models,
- what are limitations of the proposed model in maintaining the temporal consistency of compute depth maps, since monocular depth model usually suffer with the flickering noises in depth computations
- Can the proposed QSCA be robust to handle the flickering noises at the boundaries of the objects
- can the authors provide visualization of point cloud using the compute depths of the original teacher model and quantized QCSA model. This helps reader to understand quality of depth, since in most use cases depth maps are in downstream task to compute point clouds.

---

> ### Author Rebuttal · Authors · 2025-07-30
>
> >**Q1**: Can the proposed QSCA method work on video depth models,
>
> **A1**: Our current work being centered on the monocular depth estimation for single RGB images, so we have not yet applied QSCA to video-based models. Additionally, short-term experimental application is challenging due to the additional complexity involved in temporal modeling and evaluation. However, the underlying principles of video depth models are fundamentally analogous to their single-image counterparts, despite differences in architecture or temporal processing. Therefore, we anticipate that QSCA can be effectively integrated into video depth models to enhance their quantization performance, making this a promising direction for future research.
>
> >**Q2**: what are limitations of the proposed model in maintaining the temporal consistency of compute depth maps, since monocular depth model usually suffer with the flickering noises in depth computations
>
> >**Q3**: Can the proposed QSCA be robust to handle the flickering noises at the boundaries of the objects
>
> **A2-3:** We sincerely thank the reviewer for raising important questions regarding the limitations of the proposed QSCA framework to video depth models, including its limitations in maintaining temporal consistency and robustness against flickering artifacts, particularly at object boundaries. QSCA is designed for post-training quantization of monocular depth estimation models that operate on single, static images, and it does not incorporate temporal modeling that accounts for inter-frame dynamics.
>
> Therefore, our study does not directly address temporal artifacts such as flickering that may arise when the model is applied to video sequences. These issues are typically handled using dedicated temporal consistency modules or training strategies that are aware of sequential information, which lie outside the scope of our quantization optimization objectives. We believe that extending or integrating QSCA with techniques that promote temporal stability would be a promising direction for future work, especially for applications in video-based environments.
>
> >**Q4**: can the authors provide visualization of point cloud using the compute depths of the original teacher model and quantized QCSA model. This helps reader to understand quality of depth, since in most use cases depth maps are in downstream task to compute point clouds.
>
> **A4:** To help illustrate the qualitative differences between the original teacher model and the quantized QSCA model, we plan to include such visualizations as part of Figure 3 in the revised manuscript. These will help demonstrate how well the quantized model preserves geometric structures in downstream applications. As the current review platform does not support image attachments, we are unable to include the figure at this stage.

---

> > ### Comment · Reviewer_89H7 · 2025-08-07
> >
> > Thankyou Authors for providing rebuttal. After reading the reviews from other reviewers and authors responses, for now I am maintaining my score. I will revisit and finalize my score after considering the perspectives of the other reviewers and seeing whether there are additional points that might warrant a further adjustment.

---

> > > ### Author Response · Authors · 2025-08-08
> > >
> > > Dear Reviewer 89H7,
> > >
> > > Thank you for your thorough review and valuable time evaluating our work. We appreciate your careful review and willingness to revisit the score after further discussions. If there is any additional clarification or experimental result that could help address remaining concerns, we would be happy to provide it promptly.
> > >
> > > Sincerely,
> > >
> > > Authors

---

### Official Review · Reviewer_aQPB · 2025-07-05

**Clarity:** 3
**Significance:** 3
**Originality:** 3
**Rating:** 4
**Confidence:** 3

**Summary:**

This paper proposes QSCA, a post-training quantization (PTQ) framework for monocular depth estimation (MDE) models, targeting 4-bit quantization of large-scale foundation models like Depth Anything. The key innovation is the insertion of self-compensating auxiliary (SCA) modules into transformer encoder and decoder blocks, trained via self-supervised distillation from the full-precision model. QSCA aims to recover accuracy degraded by low-bit quantization without requiring labeled data. Experimental results across multiple benchmarks demonstrate improved δ₁ and AbsRel metrics over standard PTQ baselines under both 4/4 and 4/6 bit-width configurations.

**Questions:**

1. What are the concrete design choices for the SCA modules (e.g., number of layers, activation functions)? Have you validated simpler alternatives?

2. Can QSCA generalize to CNN-based MDE models (e.g., Monodepth2, BTS), or is it specific to ViT-based architectures?

3. Why are recent lightweight QAT/optimization-based quantization methods (e.g., PTQ4ViT, Adaround++) excluded from comparison?

4. How robust is the proposed method under noisy or poorly calibrated input distributions (e.g., under domain shift or low-light scenes)?

5. Would performance still hold if full-precision model outputs are not available (e.g., in proprietary foundation models)?

**Ethical Concerns:**

["NO or VERY MINOR ethics concerns only"]

**Final Justification:**

The authors have addressed my concerns raised during the review process.

**Limitations:**

The authors acknowledge the lack of comparison with reconstruction-based PTQ methods, citing GPU constraints. However, this weakens the completeness of the empirical evaluation. The proposed method is also architecture-specific (transformer + DPT), and generalization to other depth models remains untested. No experiments on actual edge hardware are provided, despite the motivation being efficiency on resource-constrained devices. Additionally, the method relies on access to full-precision model predictions, which may not be feasible in many realistic deployment cases.

**Quality:**

3

**Strengths And Weaknesses:**

Strengths:

Well-motivated problem: Efficient quantization of large MDE models is an important and under-explored problem in practical deployment scenarios.

Methodological novelty: The design of SCA modules with closed-form initialization and self-supervised training is a meaningful contribution, particularly under zero-label assumptions.

Thorough evaluation: Experiments span multiple challenging datasets and strong PTQ baselines, and results consistently favor QSCA.

Computational efficiency: QSCA achieves a notable reduction in reconstruction time compared to QDrop and BRECQ, with only minor parameter overhead.

Weaknesses:

Limited rigor in analysis: The lack of statistical significance measures (e.g., variance, error bars) is concerning, especially when making performance claims across datasets.

Lack of real deployment evidence: Despite emphasizing "resource-constrained environments," there is no evaluation on actual edge hardware (e.g., mobile devices, embedded GPUs).

SCA architecture not deeply analyzed: The design choices of the SCA modules (e.g., depth, kernel size, nonlinearity) are not ablated or justified in detail.

Limited generality: The framework is tightly coupled with transformer-based architectures (DPT), and applicability to other MDE backbones is not addressed.

Comparative scope: QSCA is only compared against classical PTQ methods. Recent hybrid PTQ/QAT or optimization-based methods are omitted, weakening the claim of state-of-the-art performance.

---

> ### Author Rebuttal · Authors · 2025-07-30
>
> >**Q1:** The lack of statistical significance measures (e.g., variance, error bars) is concerning, especially when making performance claims across datasets.
>
> **A1:** We conducted additional experiments to quantify the statistical robustness of our results. We evaluated the performance of our QSCA framework under using five independently and randomly chosen dataset for calibration and train. We then measured the δ₁ and AbsRel metrics across these runs and calculated the mean and standard deviation. Finally, we show the results of the experiment below to prove the robust performance of our method, with standard deviations consistently remaining below 0.5% in most cases, demonstrating its stability.
> |Dataset|NYU||||KITTI||||
> -|-|-|-|-|-|-|-|-
> |Model|*E.* ViT-S||*E.* ViT-B||*E.* ViT-S||*E.* ViT-B|
> W/A|δ₁↑|AbsRel↓|δ₁↑|AbsRel↓|δ₁↑|AbsRel↓|δ₁↑|AbsRel↓
> 4/4|0.8097±0.004|0.1377±0.001|0.6252±0.014|0.2141±0.006|0.7273±0.005|0.1874±0.003|0.6004±0.028|0.2523±0.013
> 4/6|0.9160±0.005|0.0902±0.003|0.9438±0.007|0.0743±0.004|0.8857±0.009|0.1161±0.006|0.8722±0.013|0.1174±0.006
> >**Q2:** Lack of real deployment evidence: Despite emphasizing "resource-constrained environments," there is no evaluation on actual edge hardware (e.g., mobile devices, embedded GPUs).
>
> **A2:** In this study, we focused on validating the effectiveness of the QSCA framework using fake quantization, which is adopted for simulating low-precision behavior without generating actual quantized models. Fake quantization is widely used, with the purpose of simulating the performance changes depending on the conducted quantization levels. While this approach provides reliable insights into quantization-aware performance, it does not enable direct measurement of latency or memory usage on real devices. We acknowledge this as a limitation of the current study, and plan to incorporate comprehensive on-device evaluations in future work to better assess the practical deployment aspects of our framework.
>
> To address this limitation, we additionally performed a both theoretical and empirical analysis to estimate the deployability of our framework. We calculated the model size difference with and without the SCA modules under 4-bit quantization to evaluate parameter efficiency. Furthermore, we converted the 8-bit quantized model into a TensorRT engine and measured its latency on an RTX4090 GPU. We then added the SCA modules and re-measured latency to analyze the overhead introduced by QSCA. Although these analyses are not full replacements for actual on-device measurements at 4-bit precision, they offer a practical approximation.
>
> ||Model size(MB)||Latency(ms)||Peak memory(MiB)||
> -|-|-|-|-|-|-
> |Model|*E.* ViT-S|*E.* ViT-B|*E.* ViT-S|*E.* ViT-B|*E.* ViT-S|*E.* ViT-B
> FP32|94.62|371.90|7.64|17.27|~334.8|~826.6
> (w/o) SCA|14.49|52.93|1.81|3.57|~78.4|~229.8
> (w/) SCA|16.27|60.02|1.82|3.79|~84.3|~240.4
>
> >**Q3:** What are the concrete design choices for the SCA modules (e.g., number of layers, activation functions)? Have you validated simpler alternatives?
>
> **A3:** Quantization is fundamentally a model compression technique aimed at reducing computational cost and memory usage for efficient deployment. In alignment with this goal, the design of our SCA modules was intentionally kept as simple and lightweight as possible to avoid adding significant overhead.
>
> Each SCA module consists of only a single linear or convolutional layer, with no hidden layers or nonlinear activation functions (e.g., GELU, ReLU, etc.). This minimal design ensures that the module does not significantly increase the number of model’s parameters or inference cost, while still effectively compensating for accuracy degradation caused by low-bit quantization. We believe this is the simplest possible structure for such a compensation module, and therefore, it is also hard for us to consider alternatives.
>
> >**Q4:** Can QSCA generalize to CNN-based MDE models (e.g., Monodepth2, BTS), or is it specific to ViT-based architectures?
>
> **A4:** We designed QSCA with a primary focus on MDE foundation models. Notably, Depth Anything employs an architecture, combining both a ViT encoder with a CNN decoder. Due to this structural characteristic, QSCA is not exclusively specialized for pure ViT-based models.
> The core idea of QSCA involves integrating lightweight SCA modules trained in a self-supervised manner. This approach is well-suited to the hybrid nature of Depth Anything, effectively addressing quantization challenges in both the ViT encoder and the CNN decoder. Based on this, we expect QSCA's compatibility with CNN-based components.
>
> >**Q5:** Why are recent lightweight QAT/optimization-based quantization methods (e.g., PTQ4ViT, Adaround++) excluded from comparison?
>
> **A5:** To address the reviewer’s concern regarding missing comparison, we have included PTQ4ViT, RepQ-ViT, and ERQ as recent optimization-based PTQ baselines, and QwT as a hybrid PTQ/QAT method. Despite being strong alternatives, QSCA consistently outperforms every compared method in accuracy and robustness, especially under low-bit settings, while the performance also remains stable across different evaluation datasets.
>
> However, applying QAT to MDE foundation models would require full retraining with the original dataset, which is computationally infeasible in our setting due to hardware limitations. QAT involves updating the model’s weight parameters, which introduces a significant training overhead and often exceeds deployment constraints in real-world settings. In contrast, our method only trains lightweight SCA modules while keeping the quantized model weights.
>
> |Dataset||NYU||||KITTI||||
> |-|-|-|-|-|-|-|-|-|-
> |Model||*E.* ViT-S||*E.* ViT-B||*E.* ViT-S||*E.* ViT-B|
> Method|W/A|δ₁↑|AbsRel↓|δ₁↑|AbsRel↓|δ₁↑|AbsRel↓|δ₁↑|AbsRel↓
> MinMax|4/4|0.5024|0.2728|0.1972|1.5735|0.3441|0.3770|0.2058|1.9612
> Percentile|4/4|0.6542|0.2006|0.5430|0.2522|0.4099|0.3418|0.3327|0.3876
> BRECQ|4/4|0.5395|0.2535|0.4692|0.2886|0.3522|0.3418|0.3160|0.3989
> Qdrop|4/4|0.7166|0.1742|0.5785|0.2334|0.3234|0.3719|0.3338|0.3855
> PTQ4ViT|4/4|0.5693|0.2393|0.5895|0.2294|0.4106|0.3439|0.3251|0.3923
> RepQ-ViT|4/4 |0.6639|0.1959|0.5410|0.2539|0.4159|0.3434|0.3164|0.3982
> ERQ|4/4|0.7062|0.1785|0.6126|0.2182|0.4847|0.3178|0.4241|0.3528
> QwT|4/4|0.8007|0.1407|0.6486|0.2024|0.6862|0.1802|0.5867|0.2417
> Ours|4/4|0.8097±0.004|0.1377±0.001|0.6252±0.014|0.2141±0.006|0.7273±0.005|0.1874±0.003|0.6004±0.028|0.2523±0.013
> |||||||||
> MinMax|4/6|0.5632|0.2417|0.4973|0.2748|0.4467|0.3290|0.3586|0.3769
> Percentile|4/6|0.8837|0.1050|0.9071|0.0958|0.8558|0.1269|0.7580|0.1687
> BRECQ|4/6|0.5786|0.2337|0.5542|0.2462|0.5223|0.2851|0.4035|0.3529
> Qdrop|4/6|0.6369|0.2071|0.6987|0.1823|0.6052|0.2408|0.5897|0.2611
> PTQ4ViT|4/6|0.6187|0.2166|0.6122|0.2201|0.4446|0.3230|0.3488|0.3848
> RepQ-ViT|4/6|0.8256|0.1304|0.8867|0.1058|0.7888|0.1502|0.5610|0.2602
> ERQ|4/6|0.9075|0.0945|0.9390|0.0766|0.8629|0.1244|0.8705|0.1133
> QwT|4/6|0.9049|0.0956|0.9089|0.0928|0.8844|0.1152|0.7292|0.1759
> Ours|4/6|0.9160±0.005|0.0902±0.003|0.9438±0.007|0.0743±0.004|0.8857±0.009|0.1161±0.006|0.8722±0.013|0.1174±0.006
>
> >**Q6 :** How robust is the proposed method under noisy or poorly calibrated input distributions (e.g., under domain shift or low-light scenes)?
>
> **A6:** To evaluate this, we conducted a cross-domain calibration experiment, where we tested the NYUv2 dataset using calibration and self-supervised training data from KITTI, and vice versa. This setup simulates domain shift scenarios where the calibration distribution differs significantly from the test distribution. The results showed noticeable performance degradation in both cases.
>
> We believe this is primarily due to the substantial differences in scene characteristics and depth distribution between NYUv2 (indoor scenes with depth range up to 10 meters) and KITTI (outdoor scenes with depth range up to 80 meters). These discrepancies pose a significant challenge for calibration and adaptation, especially in a post-training setting without access to labeled data. Therefore, enhancing the robustness of this self-supervised adaptation for severe domain shifts is a promising and important avenue for our future research.
>
> |Test dataset:|NYU|||||
> -|-|-|-|-|-
> |Model||*E.* ViT-S||*E.* ViT-B|
> W/A|Calibration & Train data|δ₁↑|AbsRel↓|δ₁↑|AbsRel↓
> 4/4|NYU|0.8097±0.004|0.1377±0.001|0.6252±0.014|0.2141±0.006
> 4/4|KITTI|0.7378±0.002|0.1668±0.001|0.5859±0.013|0.2360±0.008
> 4/6|NYU|0.9160±0.005|0.0902±0.003|0.9438±0.007|0.0743±0.004
> 4/6|KITTI|0.8899±0.005|0.1058±0.003|0.9261±0.005|0.0867±0.005
>
> |Test dataset:|KITTI|||||
> -|-|-|-|-|-
> |Model||*E.* ViT-S||*E.* ViT-B|
> W/A|Calibration & Train data|δ₁↑|AbsRel↓|δ₁↑|AbsRel↓
> 4/4| KITTI|0.7273±0.005|0.1874±0.003|0.6004±0.028|0.2523±0.013
> 4/4| NYU|0.4683±0.010|0.3043±0.004|0.4199±0.028|0.3246±0.004
> 4/6| KITTI|0.8857±0.009|0.1161±0.006|0.8722±0.013|0.1174±0.006
> 4/6| NYU|0.5342±0.002|0.2725±0.002|0.5196±0.008|0.2807±0.005
>
> >**Q7:** Would performance still hold if full-precision model outputs are not available (e.g., in proprietary foundation models)?
>
> **A7:** Our QSCA workflow follows the usual conventions of PTQ, where the full precision model is accessed only once during the preparation stage. Afterwards, every inference step relies solely on the quantized model. In the quantization preparation phase, the FP model serves two roles. It first provides the statistics needed to set each layer’s scale and zero point, and secondly to generate reference outputs that guide the compensation module training. This matches the assumption adopted by widely used PTQ baselines such as BRECQ and QDrop, which likewise depend on full precision outputs for reconstruction. When the quantization is finished, QSCA runs entirely with low precision weights and activations, and it no longer interacts with the FP model. Because our study is based around this standard PTQ setting, we did not consider scenarios in which the FP model is inaccessible, such as proprietary foundation models.

---

> > ### Comment · Reviewer_aQPB · 2025-08-06
> >
> > The authors have addressed my concerns.

---

> > > ### Author Response · Authors · 2025-08-08
> > >
> > > Dear Reviewer aQPB,
> > >
> > > We sincerely appreciate your recognition that all concerns have been addressed.
> > > Your constructive feedback throughout the review process has been invaluable in refining our work and we are grateful for the time and effort you have dedicated to evaluating our submission.
> > >
> > > Sincerely,
> > >
> > > Authors

---

### Note · Authors · 2025-08-12

Dear Area Chair and Reviewers,

We want to provide Final Remarks on our rebuttal and discussion. We believe the insightful feedback from the reviewers has significantly strengthened our paper.

### **Summary of Rebuttal Responses**

We addressed several key concerns raised by the reviewers:

1. Statistical Significance: Reviewers aQPB and hdHX pointed out the lack of statistical metrics. We conducted experiments using different random seeds and confirmed the robustness of our results, typically showing a standard deviation of less than 0.5%.

2. SCA Module Design: Reviewers aQPB and hdHX requested more detailed explanations on the design of our SCA module. We clarified that to achieve a closed-form initialization, the optimal design is to connect a single linear SCA to the ViT encoder and a conv SCA to the CNN decoder.

3. Efficiency Metrics: Reviewers aQPB, j8e1, and hdHX noted the absence of efficiency comparisons (e.g., latency, GPU memory). We acknowledged that our method primarily uses fake quantization. To address these concerns, we provided model size comparisons and presented potential efficiency gains by measuring latency and peak memory consumption using TensorRT engine.

4. Comparative Analysis: Reviewers aQPB, j8e1, and hdHX requested comparisons with more recent PTQ methods and QwT. We expanded our experimental section to include comparisons with RepQ-ViT, ERQ, and QwT at 4-bit precision. Additionally, per reviewer j8e1's suggestion, we included an 8-bit comparison to demonstrate the robustness of our method across different precisions.

5. Robustness to Video Models: Reviewer 89H7 inquired about the applicability of our method to video depth models. Although our current work focuses on single-image depth models, we argued that the core operations are similar. Therefore, we anticipate that QSCA can be effectively integrated into video depth models to enhance their quantization performance.


### **Resolution of Concerns**

Reviewers aQPB and hdHX have confirmed that their concerns have been addressed. Reviewer 89H7 indicated that their final decision would be made after considering the feedback from other reviewers. While we have not received a direct response from reviewer j8e1, we are confident that our responses have addressed their concerns.

We are grateful for the constructive feedback from all the reviewers, which has been invaluable in improving the quality and clarity of our paper.

Sincerely,

Authors

---

### Decision · Program_Chairs · 2025-09-17

**Decision:**

Accept (poster)

**Comment:**

The final ratings are 2 borderline accepts, 2 accepts. The AC have read the reviews and rebuttal, and discussed the submission with the reviewers. The reviewers raised a number of points during the review phase including statistical analysis in experiments, computational efficiency of method, clarification of proposed design, and additional experiments, ablation studies, and comparisons. The authors were able to address most of these points during the rebuttal and discussion phases with extensive experiments. Overall, the reviewers were swayed from the initial ratings to a positive consensus. The AC notes that many of the experiments were critical and should have been considered in the original submission. The authors should incorporate the feedback and suggestions provided by the reviewers, and the materials presented in the rebuttal, which would improve the next revision of the manuscript.